# Neighborhood and Global Perturbations Supported SAM in Federated Learning: From Local Tweaks To Global Awareness

## Abstract

Federated Learning (FL) can be coordinated under the orchestration of a central server to build a privacy-preserving model without collaborative data exchange. However, participant data heterogeneity leads to local optima divergence, affecting convergence outcomes. Recent research focused on global sharpness-aware minimization (SAM) and dynamic regularization to enhance consistency between global and local generalization and optimization objectives. Nonetheless, the estimation of global SAM introduces additional computational and memory overhead. At the same time, the local dynamic regularizer cannot capture the global update state due to training isolation. This paper proposes a novel FL algorithm, FedTOGA, designed to consider optimization and generalization objectives while maintaining minimal uplink communication overhead. By linking local perturbations to global updates, global generalization consistency is improved. Additionally, linking the local dynamic regularizer to global updates increases the perception of the global gradient and enhances optimization consistency. Global updates are passively received by clients, reducing overhead. We also propose neighborhood perturbation to approximate local perturbation, analyzing its strengths and working principle. Theoretical analysis shows FedTOGA achieves faster convergence $O(1/T)$ on the non-convex function. Empirical studies demonstrate that FedTOGA outperforms existing algorithms, with a 1% accuracy increase and 30% faster convergence, achieving state-of-the-art.

## 1 Introduction

The widespread connectivity of mobile terminals has dramatically propelled the development of industries related to big data. However, the massive data throughput has led to communication link congestion and increased privacy risks. Consequently, to safeguard data privatization and localization, FL McMahan et al. (2017) has garnered significant attention as a distributed machine learning (ML) method that avoids the need for data exchange. Nonetheless, due to the variations in data distribution among participants Fan et al. (2022; 2024c), conflicts in local optimization targets arise, potentially causing the global loss function to converge to a sharp local minimum Woodworth et al. (2020). As illustrated in Figures 1a-1c, with an increase in local heterogeneity, there is a steep rise in the sharpness of the global model loss. Moreover, due to the limitations of uplink bandwidth to the global server Speedtest (2024), FL employs a "Computation-Then-Aggregation" (CTA) strategy Zhang et al. (2020), which utilizes multiple rounds of local training and partial participation to alleviate communication bottlenecks. However, by increasing synchronization intervals and reducing participation rates, the discrepancy between local and global models will be significantly amplified Wang et al. (2020); Li et al. (2020b).

In response to these challenges, most studies address global consistency issues via the Empirical Risk Minimization (ERM) Malinovsky et al. (2020). However, when handling highly heterogeneous datasets, global solutions may become trapped in steep local minima, rendering it difficult to provide reliable estimates Sun et al. (2023b) and potentially causing the optimizer to stagnate. Consequently, recent innovations have leveraged Sharpness-Aware Minimization (SAM) Foret et al. (2021), which seeks to identify a flatter minimum by minimizing the perturbed loss of the model, thereby enhancing generalization capabilities. FedSAM was introduced by incorporating SAM into

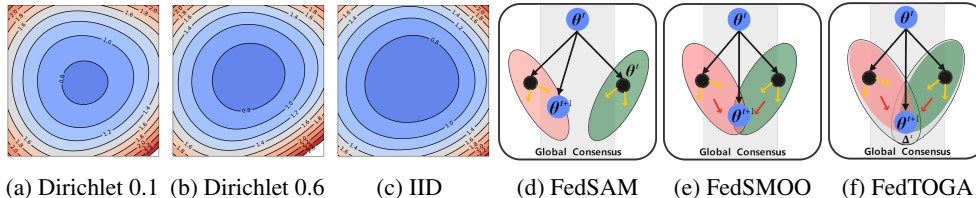

(a) Dirichlet 0.1  (b) Dirichlet 0.6  (c) IID  (d) FedSAM  (e) FedSMOO  (f) FedTOGA

Figure 1: Fig.(a)-(c) shows the loss surface under FL IID and the Non-IID setting and Fig.(d)-(f) shows the FL system, where the gray color represents the global consensus and the colored regions represent the local knowledge. In Fig.(d), no further consensus can be increased in FL only supported by the SAM. In Fig.(e), a dynamic regularizer is introduced in some work to increase global generalization. In Fig.(f), we further introduce Global Update to extend the generalization.

FL Qu et al. (2022), and further, momentum-based algorithms were integrated, resulting in the proposal of MoFedSAM. FedGAMMA Dai et al. (2023a) replaced ERM with SAM in Scaffold to enhance its performance and alleviate model bias. FedSpeed Sun et al. (2023c) integrated FedDyn into FedSAM to bolster performance. However, the approach based on minimizing local sharpness loss fails to capture the flatness of the global loss surface, as depicted in Fig.1a; localized knowledge does not allow for effective consensus. Therefore, FedSMOO Sun et al. (2023b) enhances local target consistency through a dynamic regulariser, as shown in Fig.1e. Furthermore, FedLESAM Fan et al. (2024a) estimates the global perturbation as the difference between the locally stored historical model from the activation round and the global model received in the current round, thereby avoiding extra computational costs. Nonetheless, FedSMOO introduces additional communication overhead and storage requirements, which can be unacceptable in real-world environments with limited bandwidth. In FedLESAM Fan et al. (2024a), the global perturbation estimate does not encompass additional local gradient ascent computations, while reducing computational overhead may lead to insufficient local generalization. In both methods, the difference in local perturbation scales may be significant when facing clients with prolonged disconnections, thereby disrupting generalization. Apart from addressing the consistency of perturbation generalization, local objective optimization has also been intensely studied, as seen in dynamic regularize algorithms Zhang et al. (2020); Acar et al. (2021); Sun et al. (2023b). However, its performance degrades significantly as the local interval expands. This is because the CTA strategy makes local fail to capture the global updates state.

To achieve a reliable, stable, and consistent global model, we propose a novel algorithm named FedTOGA, as illustrated in Fig.1f. FedTOGA initially guides global update gradients to merge with local perturbations, thereby enhancing local generalization consistency. Simultaneously, it employs global updates to correct the local dynamic regularizer, reinforcing consistency with the global optimization objective. This approach significantly improves performance, even under extreme conditions characterized by highly heterogeneous data or limited client participation. Due to the communication interval, the universal SAM optimizer applied on the global server cannot precisely capture the perturbations occurring during local updates on client devices. We introduce the global update gradient as an approximation to maintain local consistency, replacing the global perturbation estimation used in FedSMOO Sun et al. (2023b). Furthermore, the universal dynamic regularizer dynamically adjusts the optimization direction based on the client drift estimated from dual variable statistics; however, it neglects the global gradient's changing trends. Hence, we consider the global stationary condition and propose leveraging the global update gradient to correct the local dynamic regularizer, further aligning with global and local objectives. At the same time, to further reduce the computational overhead, we propose the neighborhood gradient perturbation. When the interval between local training sessions on the client side exceeds one, the client simulates the current perturbation by utilizing cached gradients stored in a gradient register, thereby alleviating computational costs. Unlike FedCM Xu et al. (2021) and MoFedSAM Qu et al. (2022), the global update is not treated as a trade-off term with the local perturbation, meaning that their coefficients do not sum to one. As the active local clients converge, they ultimately reach a globally stationary state characterized by a smooth loss landscape.

Theoretically, FedTOGA can achieve a rapid convergence rate of $O(1/T)$ in non-convex settings. Extensive evaluations were conducted on the CIFAR10/100 datasets, demonstrating that FedTOGA

achieves faster convergence rates and higher generalization accuracy in practice. These results were obtained in comparison to 17 baseline methods, including FedAvg, FedAdam, FedYogi, SCAFFOLD, FedACG, FedCM, FedDyn, FedDC, FedRCL, FedSAM, MoFedSAM, FedGAMMA, FedSMOO, FedSpeed, FedLESAM, FedLESAM-D, and FedLESAM-S.

- We propose a novel FL algorithm, FedTOGA, the first global perturbation technique that uses a **merged global update**, and the first local dynamic regularizer that **employs the global update**. This method effectively reduces uplink communication overhead, ensuring rapid convergence and strong generalization.

- We introduce the concept of neighborhood perturbation to mitigate local computation and enhance generalization for the first time. This approach integrates or substitutes local perturbation by leveraging gradient registers without incurring additional overhead. We further analyze its benefits and working principles.

- We provide a theoretical convergence rate analysis, demonstrating that FedTOGA attains a rapid $O(1/T)$ convergence rate in non-convex settings. Additionally, we performed extensive empirical evaluations on the CIFAR10 and CIFAR100 datasets using various neural network architectures to validate the superior performance of FedTOGA, particularly in scenarios involving highly heterogeneous and sparse participants, where it significantly outperformed existing methods.

## 2 PRELIMINARIES

This section shows the preliminaries of FL and SAM, and related works are in Appendix A.1.

**Federated Learning** The goal of the FL framework is to build global models that minimize the average experience loss of participating clients:

$$\arg\min_{\theta} f(\theta) = \frac{1}{N} \sum_{i \in N} f_i(\theta); \quad f_i(\theta) \triangleq \mathbb{E}_{\xi_i \sim D_i} f_i(\theta, \xi_i). \tag{1}$$

Where $f : \mathbb{R} \to \mathbb{R}^d$ is denoted as the global objective function, $\theta$ is a model parameter, $N$ is the total number of all the participating clients, and $\xi_i$ is a randomly sampled data from the distribution $D_i$ subject to data heterogeneity. $f_i$ is the loss function for the $i$-th client.

**Sharpness Aware Minimization** Many studies Hochreiter & Schmidhuber (1994); Dinh et al. (2017) have pointed out that a flat minimum implies a better generalization performance, which possesses greater robustness to model perturbations. To minimize sharpness, Keskar et al. (2017); Foret et al. (2021) proposed SAM:

$$\arg\min_{\theta} \left\{ f_{sam}(\theta) = \arg\max_{\|\delta\| \leq \rho} f(\theta + \delta) \right\}. \tag{2}$$

SAM extends the search by a one-step gradient ascent perturbation and a one-step gradient descent to reduce sharpness and loss. First, calculate the gradient ascent perturbation $\delta = \rho \frac{\nabla f(\theta)}{\|\nabla f(\theta)\|}$. $\rho$ is the perturbation learning rate. The gradient is then computed after adding the perturbation using the model and updating the model $\tilde{g} = \nabla f(\theta + \delta); \theta = \theta - \eta\tilde{g}$, where $\eta$ is the learning rate.

### 2.1 RETHINK FEDSAM AND OTHERS

The limitations of SAM in FL systems Qu et al. (2022) have been widely discussed Sun et al. (2023b); Fan et al. (2024a); Lee et al. (2024), with the main conflict coming mainly from centralized training vs. Distributed Computing Perturbation Differences. The centralized SAM Qu et al. (2022) training objectives are as follows:

$$\max_{\|\delta\| \leq \rho} \mathbb{E}_{\xi \sim D} f(\theta + \delta, \xi) = \max_{\|\delta\| \leq \rho} \mathbb{E}_i \mathbb{E}_{\xi_i \sim D_i} f(\theta + \delta, \xi_i). \tag{3}$$

where $D = \mathbb{E}_i D_i$, some work applies SAM directly to the FL paradigm Sun et al. (2023c); Dai et al. (2023a), and reformulates its goal as follows:

$$\max_{\|\delta_i\| \leq \rho} \mathbb{E}_i \mathbb{E}_{\xi_i \sim D_i} f_i(\theta_i + \delta_i, \xi_i). \tag{4}$$

where the model $\theta$, and the perturbation $\delta$ are isolated due to the CTA of FL, and the $\theta_i$ represents the local model. In this case, minimizing local sharpness in isolation does not effectively achieve a global flat minimum. As a result, maintaining consistency between the global and client models becomes more difficult as the local update interval and the degree of data heterogeneity increase Fan et al. (2024a).

Some recent studies, FedSAM Caldarola et al. (2022), MoFedSAM Qu et al. (2022), FedGAMMA Dai et al. (2023a), FedSpeed Sun et al. (2023c) have not resolved the internal perturbation variance contradiction. MoFedSAM uses momentum to weigh the perturbation gradient against the global gradient to alleviate this problem, FedGAMMA Dai et al. (2023a) uses variance reduction techniques, and FedSpeed Sun et al. (2023c) uses dynamic regularization to alleviate this contradiction. FedSMOO Sun et al. (2023b) notices this contradiction for the first time and uses dynamic regularization to correct the discrepancy between local and global perturbations, FedSOL Lee et al. (2024) employs perturbation orthogonality to find a consistent direction of perturbation, FedLESAM Fan et al. (2024a) believes that computing the perturbations requires additional computation and therefore opens up additional storage locally to approximate the estimated global perturbations. However, FedSMOO introduces additional computation, which increases the overhead of the clients in FL. As the set of activated clients $S_t$ decreases sharply, the perturbation estimation by FedLESAM is more affected. For a more detailed description of the limitations, see Appendix A.2.

## 2.2 RETHINK DYNAMIC REGULARIZER IN FL

Dynamic regularisation is intensively studied in FL(FedPD/FedDyn/FedSMOO) Wang et al. (2022); Gong et al. (2022); Acar et al. (2021); Sun et al. (2023b), mainly used to correct local optimization biases. Consider a standard edge Augmented Lagrangian(AL) function in dynamic regularisation:

$$F_{fed} : \frac{1}{N} \sum_{i \in N} \left\{ f_i + \left\langle h_i^t, \theta^t - \theta_i^t \right\rangle + \frac{1}{2\alpha} \|\theta^t - \theta_i^t\|^2 \right\}. \tag{5}$$

The dual variable $h_i$ can be interpreted as a signed "correction vector" (positive or negative), quantifying the discrepancy between $\theta_i^t$ and $\theta^t$ and providing the direction for adjustments in optimization Zhang et al. (2020); Gong et al. (2022). Consider the first-order condition $\nabla f_i(\theta_i^t) - h_i^t + \frac{1}{\alpha}(\theta_i^t - \theta^t) = 0$, which ensured that convergence to a stationary point at each training iteration. And consider the update rule of the dual variable $h_i = h_i - \frac{1}{\alpha}(\theta_{i,K}^t - \theta_{i,0}^t)$, we have $\nabla f_i(\theta_i^t) - \nabla f_i(\theta_i^{t-1}) + \frac{1}{\alpha}(\theta_i^t - \theta^t) = 0$. When $t \to \infty$, have $\theta_i^\infty \to \theta_i^{\infty-1}$, then $\theta_i^\infty \to \theta^\infty$ Acar et al. (2021); Sun et al. (2023b). However, each subproblem's stationary points typically differ, particularly in the FL setting with heterogeneous data distributions. Observing the stationary condition in problem 1, $\sum_{i \in N} \nabla f_i(\theta^*) = 0$, it is implied that any given $-\nabla f_i(\theta^*)$ could be partially or fully offset by $\nabla f_j(\theta^*)$, where $i \neq j$. Consequently, existing studies that focus solely on local correction are insufficient; there is a need to incorporate further considerations of the global stationary condition within the local optimization process.

## 3 MOTIVATION

Therefore, we consider three questions:

1. How can we efficiently estimate global perturbations without adding extra overhead?

2. How can we reduce computational overheads and enhance local generalization?

3. How can we further align local and global objectives?

## 4 METHODOLOGY

### 4.1 ESTIMATE GLOBAL PERTURBATION

As mentioned above, we aim to efficiently estimate each client's global perturbations(G-perturbations) without incurring additional storage or computational overhead. To achieve this, we

---

**Algorithm 1:** FedTOGA Algorithm

1   Initial model parameters $\theta^0$, initial global update $\Delta^{-1}$, local dual variable $h_i$, global dual variable $h$, local perturbation gradient $\tilde{g}_{i,-1}$, total communication rounds $T$, penalized coefficient for the quadratic term $\alpha$, Correction coefficient for perturbation and dual term $\kappa, \beta$

2   **for** *each round $t \in [T] \triangleq \{0, 1, 2, \cdots, T-1\}$* **do**

3      Sample the active client set $S_t \subseteq [N]$.

4      **for** $i \in S_t$ *in parallel* **do**

5         $\theta_i^{t+1} \leftarrow$ **Client Update**$(\theta^t, \Delta^t)$;     **communicate** $\theta_i^t$ to server ;

6      **end**

7      $\Delta^{t+1} = -\frac{1}{MK}\sum_{i \in S_t}(\theta_i^{t+1} - \theta^t)$; $h^{t+1} = h^t - \frac{1}{\alpha}K\Delta^{t+1}$; $\theta^{t+1} = \frac{1}{M}\sum_{i \in S_t}\theta_i^{t+1} - \alpha h^{t+1}$

8   **end**

9   **Client Update**$(\theta_t, \Delta_t)$: $\theta_{i,0}^t = \theta^t$

10   **for** *local epoch $k \in [K] \triangleq \{0, 1, 2, \cdots, K-1\}$* **do**

11      sample a mini-batch data $\xi_{i,k}^t$; $g_{i,k}^t = \nabla f(\theta_{i,k}^t; \xi_{i,k}^t)$;Perturbation: $\delta_{i,k}^t = \rho \frac{g_{i,k}^t + \kappa\Delta^t}{\|g_{i,k}^t + \kappa\Delta^t\|}$

12      extra-step: $\tilde{g}_{i,k}^t = \nabla f_i(\theta_{i,k}^t + \delta_{i,k}^t; \xi_{i,k}^t)$; $\theta_{i,k+1}^t = \theta_{i,k}^t - \eta_l\left(\tilde{g}_{i,k}^t - h_i^t + \frac{1}{\alpha}\left(\theta_{i,k}^t - \theta_{i,0}^t\right) + \beta\Delta^t\right)$

13   **end**

14   $h_i^{t+1} = h_i^t - \frac{1}{\alpha}\left(\theta_{i,K}^t - \theta_{i,0}^t\right)$;    **return** $\theta_i^{t+1} = \theta_{i,K}^t$

---

first recall the definition of sharpness-aware minimization in FL:

$$\min_\theta \left\{ f = \frac{1}{N}\sum_{i \in N} \max_{\|\delta_i\| \le \rho} \mathbb{E}_i \mathbb{E}_{\xi_i \sim D_i} f_i(\theta_i + \delta_i, \xi_i) \right\} \tag{6}$$

Therefore, we can obtain that at $t$ round, $k$ moments, the virtual global perturbation variable $\delta_k^t = \rho \frac{\nabla f(\theta^t)}{\|\nabla f(\theta^t)\|} = \rho \frac{\sum_{i \in N}\nabla f_i(\theta_k^t)}{\|\sum_{i \in N}\nabla f_i(\theta_k^t)\|} \approx \rho \frac{\sum_{i \in S}\nabla f_i(\theta_k^t)}{\|\sum_{i \in S}\nabla f_i(\theta_k^t)\|}$. The $\theta_k^t$ denotes the global model at virtual moment $k$, which is computed as $\theta_k^t = \frac{1}{M}\sum_{i \in S_t}\theta_{i,k}^t$. However, due to the CTA strategy in the FL paradigm, the set of clients does not have effective access to the global model $\theta_k^t$ at each moment in time. Thus, the global perturbation $\delta$ cannot be computed correctly. Inspired by the FedCMXu et al. (2021) strategy, we estimate the global update $\Delta^t \approx \nabla f(\theta^t)$ by passing the global update variable. Finally, we define the update strategy for the global perturbation SAM of FedTOGA as follows: $\delta_k^t = \rho \frac{\nabla f(\theta^t)}{\|\nabla f(\theta^t)\|} \approx \rho \frac{g_{i,k}^t + \kappa\Delta^t}{\|g_{i,k}^t + \kappa\Delta^t\|}$; $\theta_{i,k}^t = \theta_{i,k-1}^t - \eta_l \nabla F_i(\theta_{i,k}^t + \rho\delta_k^t)$. The differences between the FedTOGA perturbation strategy and similar works can be viewed in Appendix A.2 Tab.5.

### 4.2   UTILIZE NEIGHBOURHOOD PERTURBATION

Besides, as stated by Fan et al. (2024a), local perturbations require additional gradient ascent computation, which may consume additional computational overhead. Therefore, how can we estimate the local perturbation without utilizing additional computation? We propose neighborhood perturbation(N-perturbation) for the first time. Specifically, when the client's local iteration interval exceeds one, the local perturbation gradient $\tilde{g}_{i,k-1}^t$ will be recorded by the cache without opening additional storage space. We can get $g_{i,k} \approx \tilde{g}_{i,k-1}^t$. We can further replace the perturbation term in the local SAM optimization and get: $\delta_{i,k}^t = \rho \frac{\tilde{g}_{i,k-1}^t + \kappa\Delta^t}{\|\tilde{g}_{i,k-1}^t + \kappa\Delta^t\|}$. This operation allows approximate estimation of local perturbations in environments with scarce client-side resources.

**Perturbation Fusion?** In the FL paradigm, the edge client SAM only captures the sharpness of a specific small batch of data, which is mitigated by the G-Perturbation technique described above to enhance generalization. Let's consider whether N-Perturbation may bring additional benefits and alleviate computational overhead. Similar to LookAhead Zhang et al. (2019), it backtracks by perturbing ascent after each gradient descent. Then, our perturbation calculation can be rewritten: $\delta_{i,k}^t = \rho \frac{g_{i,k} + \tilde{g}_{i,k-1}^t + \kappa\Delta^t}{\|g_{i,k} + \tilde{g}_{i,k-1}^t + \kappa\Delta^t\|}$. Appendix A.3 provides a more in-depth discussion of N-perturbation.

### 4.3 GLOBAL CORRECTION IN DYNAMIC REGULARIZER

To effectively avoid performance degradation and further improve the optimization objective consistency, we also adopt dynamic regularization Acar et al. (2021) that merges the global update $\Delta$ correction on each local client and takes the form of an ADMM-like method on the server to minimize the global objective $f$ effectivel Zhang et al. (2020). This is the first dynamic regularisation FL framework that considers merging the global update. First, we consider the global Augmented Lagrangian(AL) function $f_{fed}$ which introduces a penalty term $\theta = \theta_i$ constraint as:

$$F_{fed} : \frac{1}{N} \sum_{i \in N} \left\{ f_i + \left\langle h_i^t, \theta^t - \theta_i^t \right\rangle + \frac{1}{2\alpha} \| \theta^t - \theta_i^t \|^2 \right\}. \tag{7}$$

In general, we can split the finite sum problem to each local client and minimize the local parameters $\theta_i$ in the AL function in each subproblem:

$$\theta_{i,K}^t = \min_{\theta_i} \left\{ f_i - \left\langle h_i^t, \theta_i^t \right\rangle + \frac{1}{2\alpha} \| \theta^t - \theta_i^t \|^2 \right\}. \tag{8}$$

Where the global dual variable $h$ is updated at each communication, and the local dual variable $h_i$ is stored locally. As stated in Sec.2.2, although recording $h_i$ helps to mitigate the local target point offset, it ignores the global gradient trend, which was not addressed in previous studies Acar et al. (2021); Sun et al. (2023b). Therefore, we use the $\Delta^t$ to approximate the global update trend. Specifically, we cause the local dual variables by adding corrections and obtaining $h_i - \beta \Delta^t$. Again, to not affect the original SAM, we use SGD to solve this problem Sun et al. (2023b). We then update the dual variables locally $h_i^{t+1} = h_i^t - \frac{1}{\alpha} \left( \theta_{i,K}^t - \theta_{i,0}^t \right)$. After finishing the local training, update $\theta^t$ to $\theta^{t+1}$ by solving the equation 7 and start the next iteration.

### 4.4 OVERVIEW OF FEDTOGA

Algorithm1 shows the detailed flow of FedTOGA. First, initialize the server-side global model $\theta$. In the global synchronization round $t$, a set $S_t$ containing $M$ clients is randomly selected from all clients $N$, and the global model $\theta_t$ is sent to the set of authorized clients $S_t$ with the global update $\Delta^t$ of the $t - 1$ round. The client first computes the original gradient $g_{i,k}^t$ according to Line.11 of the algorithm and subsequently, computes the SAM gradient $\delta_{i,k}^t$ corrected by $\Delta^t$ in Line.11, with the neighborhood perturbation variable $\tilde{g}_i$ being optional. In Line.12, we use the formula 8 for local dual variable correction to update the local model $\theta_i$ and update the local dual variable $h_i$ via Line.14. After local training, FedTOGA sends only $\theta_i^t$ to the server for aggregation. In line 7 of the algorithm, the server updates the global model from $\theta^t$ to $\theta^{t+1}$ by minimizing the function 7. This process is repeated until $T$-1.

## 5 THEORETICAL ANALYSIS

**Assumption 1.** *The loss function $f_i$ is L-Smooth, i.e., $f_i(y) - f_i(x) \leq \langle \nabla f_i(x), y - x \rangle + \frac{L}{2} \| y - x \|^2$.*

**Assumption 2.** *Unbiased and bounded variance of stochastic gradient. The stochastic gradient $\tilde{\nabla} f_i(x) = \nabla f_i(x, \xi_i)$ computed by the $i$-th client using mini-batch $\xi$ is an unbiased estimator of $\nabla f_i(x)$, i.e. $\mathbb{E}[\tilde{\nabla} f_i(x)] = \nabla f_i(x), \mathbb{E} \| \tilde{\nabla} f_i(x) - \nabla f_i(x) \|^2 \leq \sigma_l^2$.*

**Assumption 3.** *Bounded Heterogeneity, for all $x \in \mathbb{R}^d$, we establish the following inequality: $\mathbb{E} \| \nabla f_i(x) - \nabla f(x) \| \leq \sigma_g$ Besides, the variance of the unit gradient is bounded: $\mathbb{E} \| \frac{\nabla f_i(x)}{\| \nabla f_i(x) \|} - \frac{\nabla f(x)}{\| \nabla f(x) \|} \| \leq \sigma_g'$. These assumptions are also used in SAM-based FL analysis Qu et al. (2022); Fan et al. (2024a).*

**Theorem 1.** *Under Assumption 1-3, For any training interval $t$ on $i$-th client, model divergence satisfies:*

$$\| \theta_{i,k}^t - v_k^t \|^2 \leq H_i(k) \tag{9}$$

*where $H_i(\tau) \leq \frac{L^2 \rho^2 \sigma_g'^2 + \sigma_g^2}{2L^2} ((1 + 2\eta_l^2 L^2)^\tau - 1)$, $\{v^t\}$ is a virtual sequence representing the global model. More details are in the Appendix C.1.*

**Remark 1.** *The difference between the local and global models will be geometrically amplified as the local interval expands, mainly from the model perturbation error and update error, and thus, it is reasonable to enhance the consistency of optimization and generalization objective (in Sec. 4).*

**Theorem 2.** *Under Assumption 1-3. When $\eta_l \leq \min\{\frac{1}{\sqrt{2128L^2K}}, \alpha\}$, and the perturbation learning rate satisfies $\rho = O(1/\sqrt{T})$, and local interval $K > \frac{\alpha}{\eta_l}$, let $\omega = \frac{1}{2} + \beta - 2128\eta_l^2 L^2 K - \beta^2 - L\alpha\beta^2$ is a positive constant with select the suitable $\eta_l$, the auxiliary sequence $\{z^t\}$ generated by executing the Algorithm 1 satisfies:*

$$\frac{1}{T}\sum_{t=0}^{T-1}\|\nabla f(z^t)\|^2 \leq \frac{f(z^0)-f^*}{T\alpha\omega} + \frac{16\alpha^3 L^2}{T\omega}\mathbb{E}_t \left\| \frac{1}{N}\sum_{i\in N} h_i^0 \right\|^2 + \frac{112L^2\eta_l^2 K}{TN\omega}\sum_{i\in N}\mathbb{E}\|h_i^0\|^2 + \Upsilon \tag{10}$$

*where the $f^*$ is the optimal of non-convex function $f$, and the term $\Upsilon$ is:*

$$\Upsilon = \frac{1}{\omega}\left(56\eta_l^2 L^2 K(3\sigma_l^2 + 16\sigma_g^2 + 5L^2\rho^2) + L^2\rho^2\right)$$

*More details are in the Appendix C.2.*

**Remark 2.** *When we set the local learning rate $\eta_l$ to satisfy $\eta_l = O(1/K)$ and the perturbation learning rate to be $O(1/T)$, FedTOGA can achieve a fast convergence rate of $O(1/T)$ when the local interval $K$ satisfies $K = O(T)$.*

**Remark 3.** *The proof of Wang et al. (2022); Gong et al. (2022); Acar et al. (2021); Sun et al. (2023b) relies on the strict assumption that the client must approach a stationary point in each round of training, which cannot be strictly fulfilled in real FL system. Therefore, we do not consider the strict assumption of the first-order condition $\tilde{g}_i^t - h_i^t + \frac{1}{\alpha}(\theta_i^t - \theta^t) + \beta\Delta^t = 0$ of the equation (8). Inspired by Sun et al. (2023c), we relax this assumption by enlarging the local intervals, which also achieves $O(1/T)$ convergence speed Sun et al. (2023b).*

**Remark 4.** *Inspired by Sun et al. (2023c), FedTOGA can also speed up convergence by increasing the setting of the local interval $K$, which is helpful for bandwidth-constrained FL systems. However, the local perturbation learning rate $\rho$ in FedSpeed restricts the upper bound $\frac{1}{\sqrt{6}\alpha L}$, and our proof slightly relaxes the limitation so that the $\rho$ only needs to satisfy $O(1/T)$.* *We can also tighten the boundary of $\frac{1}{\omega}$ by adjusting $\beta$ appropriately.*

## 6 EXPERIMENTS

### 6.1 EXPERIMENTAL SETUPS.

**Baselines.** We compare FedTOGA with the FedAvg McMahan et al. (2017) and SAM-base FL methods, including FedSAM, MoFedSAM Qu et al. (2022), FedGAMMA Dai et al. (2023a), FedSMOO Sun et al. (2023b), FedSpeed Sun et al. (2023c), with the recent FedLESAM Fan et al. (2024a). Also, we compare with momentum-based FL algorithms, for example, FedAdam, FedYogi Reddi et al. (2021), FedACG Kim et al. (2024), FedCM Xu et al. (2021). In addition, methods based on local consistency are also in the comparison, including FedDyn Acar et al. (2021), SCAFFOLD Karimireddy et al. (2020), FedDC Gao et al. (2022) with FedRCL Seo et al. (2024).

**Experimental Details.** We adopt the same experimental setup as in Sun et al. (2023b); Fan et al. (2024a) for a fair comparison. The datasets CIFAR10 and CIFAR100 are utilized in the experiments. We follow the methodologies outlined in Dai et al. (2023b); Sun et al. (2024); Fan et al. (2024b) to simulate client data using Dirichlet and Pathological splits in non-IID scenarios. We use SGD as the optimizer, with a client learning rate $\eta_l$ set to 0.1 and a global learning rate of 1. The weight decay is fixed at $1e^{-3}$. To further assess the generalization capability of our method, we conduct experiments with two models, LeNet and ResNet18 He et al. (2016). For LeNet, the learning rate decays by 0.997 per epoch, whereas for ResNet18 He et al. (2016), it decays by 0.998. For CIFAR10, the batch size is 50, and the number of local epochs is 5. For CIFAR100, the batch size is set to 20, and the number of local epochs is set to 2. In FedTOGA, the local perturbation correction coefficient $\kappa$ is set to 1, the dual variable correction coefficient $\beta$ is set to 0.8, and the penalized coefficient $\alpha$ is set to 0.1. Consistent with several other algorithms, the perturbation magnitude $\rho$ is set to 0.1, except for FedSAM, MoFedSAM, and FedLESAM, where it is set to 0.01. Detailed information about the experimental setup can be found in Appendices B.2.

Table 1: Dirichlet coefficients $u$ selected from $\{0.1, 0.6\}$, and $c$ is the Pathological coefficient, i.e., the number of active categories in each client. The two datasets have 100 clients in the upper part with 10% active in each round and 200 clients in the lower part with 5% active in each round.(LeNet)

| Method | CIFAR10 | | | | CIFAR100 | | | |
|---|---|---|---|---|---|---|---|---|
| Partition | Dirichlet | | Pathological | | Dirichlet | | Pathological | |
| Coefficient | $u=0.6$ | $u=0.1$ | $c=6$ | $c=3$ | $u=0.6$ | $u=0.1$ | $c=20$ | $c=10$ |
| FedAvg | $80.28^{\pm0.14}$ | $74.68^{\pm0.19}$ | $80.59^{\pm0.18}$ | $78.10^{\pm0.23}$ | $47.35^{\pm0.16}$ | $45.56^{\pm0.20}$ | $46.46^{\pm0.20}$ | $43.43^{\pm0.27}$ |
| FedAdam | $80.39^{\pm0.17}$ | $71.52^{\pm0.29}$ | $81.02^{\pm0.20}$ | $77.88^{\pm0.23}$ | $48.94^{\pm0.21}$ | $43.62^{\pm0.25}$ | $44.86^{\pm0.25}$ | $41.58^{\pm0.27}$ |
| FedYogi | $80.11^{\pm0.19}$ | $73.58^{\pm0.25}$ | $81.08^{\pm0.21}$ | $78.10^{\pm0.20}$ | $48.41^{\pm0.21}$ | $45.44^{\pm0.22}$ | $46.18^{\pm0.22}$ | $42.07^{\pm0.25}$ |
| SCAFFOLD | $82.87^{\pm0.12}$ | $78.00^{\pm0.16}$ | $83.31^{\pm0.10}$ | $80.29^{\pm0.15}$ | $53.68^{\pm0.21}$ | $50.33^{\pm0.24}$ | $51.30^{\pm0.22}$ | $47.71^{\pm0.22}$ |
| FedACG | $82.87^{\pm0.14}$ | $77.51^{\pm0.16}$ | $82.86^{\pm0.12}$ | $80.84^{\pm0.17}$ | $52.88^{\pm0.20}$ | $48.72^{\pm0.23}$ | $50.24^{\pm0.21}$ | $46.08^{\pm0.24}$ |
| FedCM | $77.04^{\pm0.30}$ | $62.75^{\pm0.31}$ | $66.58^{\pm0.29}$ | $71.20^{\pm0.33}$ | $43.08^{\pm0.19}$ | $34.69^{\pm0.26}$ | $36.27^{\pm0.18}$ | $28.48^{\pm0.30}$ |
| FedDyn | $82.31^{\pm0.13}$ | $78.05^{\pm0.19}$ | $83.13^{\pm0.18}$ | $79.96^{\pm0.19}$ | $49.97^{\pm0.19}$ | $45.85^{\pm0.29}$ | $47.41^{\pm0.21}$ | $43.29^{\pm0.19}$ |
| FedDC | $83.58^{\pm0.14}$ | $78.50^{\pm0.19}$ | $84.00^{\pm0.16}$ | $81.72^{\pm0.17}$ | $51.99^{\pm0.15}$ | $48.75^{\pm0.21}$ | $49.53^{\pm0.19}$ | $44.82^{\pm0.23}$ |
| FedRCL | $77.62^{\pm0.11}$ | $68.79^{\pm0.16}$ | $78.28^{\pm0.15}$ | $76.04^{\pm0.19}$ | $46.34^{\pm0.24}$ | $42.28^{\pm0.17}$ | $44.06^{\pm0.19}$ | $39.64^{\pm0.21}$ |
| FedSAM | $81.58^{\pm0.15}$ | $77.67^{\pm0.15}$ | $82.15^{\pm0.15}$ | $79.23^{\pm0.23}$ | $48.08^{\pm0.21}$ | $46.86^{\pm0.26}$ | $46.71^{\pm0.25}$ | $43.41^{\pm0.22}$ |
| MoFedSAM | $77.17^{\pm0.12}$ | $66.24^{\pm0.15}$ | $77.44^{\pm0.15}$ | $72.15^{\pm0.19}$ | $43.30^{\pm0.18}$ | $34.43^{\pm0.21}$ | $36.50^{\pm0.19}$ | $29.92^{\pm0.24}$ |
| FedGAMMA | $83.88^{\pm0.13}$ | $78.61^{\pm0.15}$ | $83.79^{\pm0.14}$ | $79.68^{\pm0.15}$ | $53.94^{\pm0.20}$ | $49.95^{\pm0.24}$ | $51.20^{\pm0.22}$ | $48.11^{\pm0.29}$ |
| FedSMOO | $\mathbf{84.82}^{\pm0.15}$ | $\mathbf{80.06}^{\pm0.16}$ | $\mathbf{85.07}^{\pm0.17}$ | $81.26^{\pm0.19}$ | $\mathbf{56.57}^{\pm0.18}$ | $52.17^{\pm0.17}$ | $53.42^{\pm0.21}$ | $48.12^{\pm0.19}$ |
| FedSpeed | $84.14^{\pm0.15}$ | $80.16^{\pm0.16}$ | $84.74^{\pm0.14}$ | $\mathbf{82.20}^{\pm0.19}$ | $53.96^{\pm0.19}$ | $\mathbf{52.29}^{\pm0.21}$ | $53.78^{\pm0.18}$ | $\mathbf{48.33}^{\pm0.20}$ |
| FedLESAM | $80.94^{\pm0.18}$ | $77.02^{\pm0.15}$ | $81.79^{\pm0.18}$ | $78.85^{\pm0.15}$ | $48.13^{\pm0.18}$ | $46.55^{\pm0.21}$ | $46.08^{\pm0.23}$ | $43.57^{\pm0.17}$ |
| FedLESAM-D | $83.28^{\pm0.15}$ | $79.12^{\pm0.18}$ | $84.20^{\pm0.19}$ | $80.91^{\pm0.16}$ | $54.88^{\pm0.19}$ | $52.08^{\pm0.22}$ | $\mathbf{54.14}^{\pm0.19}$ | $48.28^{\pm0.22}$ |
| FedLESAM-S | $83.39^{\pm0.12}$ | $78.23^{\pm0.17}$ | $83.99^{\pm0.19}$ | $81.20^{\pm0.15}$ | $53.29^{\pm0.15}$ | $50.12^{\pm0.21}$ | $52.20^{\pm0.20}$ | $47.29^{\pm0.17}$ |
| FedTOGA(ours) | $\mathbf{86.01}^{\pm0.12}$ | $\mathbf{82.05}^{\pm0.11}$ | $\mathbf{85.71}^{\pm0.13}$ | $\mathbf{84.00}^{\pm0.12}$ | $\mathbf{57.25}^{\pm0.13}$ | $\mathbf{53.45}^{\pm0.13}$ | $\mathbf{55.49}^{\pm0.13}$ | $\mathbf{51.27}^{\pm0.18}$ |
| FedAvg | $77.53^{\pm0.17}$ | $74.60^{\pm0.23}$ | $79.21^{\pm0.25}$ | $76.20^{\pm0.23}$ | $43.86^{\pm0.21}$ | $42.70^{\pm0.24}$ | $42.94^{\pm0.25}$ | $42.28^{\pm0.29}$ |
| FedAdam | $79.39^{\pm0.19}$ | $74.49^{\pm0.31}$ | $79.53^{\pm0.23}$ | $76.09^{\pm0.25}$ | $45.34^{\pm0.25}$ | $42.79^{\pm0.23}$ | $43.57^{\pm0.25}$ | $40.66^{\pm0.29}$ |
| FedYogi | $79.95^{\pm0.21}$ | $75.29^{\pm0.25}$ | $79.73^{\pm0.22}$ | $77.64^{\pm0.23}$ | $46.67^{\pm0.25}$ | $43.02^{\pm0.24}$ | $44.70^{\pm0.27}$ | $41.33^{\pm0.30}$ |
| SCAFFOLD | $81.18^{\pm0.15}$ | $76.11^{\pm0.19}$ | $82.44^{\pm0.17}$ | $78.52^{\pm0.17}$ | $51.45^{\pm0.25}$ | $47.19^{\pm0.27}$ | $48.26^{\pm0.28}$ | $46.82^{\pm0.26}$ |
| FedACG | $82.57^{\pm0.17}$ | $78.47^{\pm0.20}$ | $82.09^{\pm0.16}$ | $80.50^{\pm0.19}$ | $51.96^{\pm0.24}$ | $49.34^{\pm0.26}$ | $50.01^{\pm0.27}$ | $46.82^{\pm0.25}$ |
| FedCM | $76.08^{\pm0.30}$ | $64.33^{\pm0.31}$ | $76.64^{\pm0.29}$ | $68.61^{\pm0.33}$ | $40.32^{\pm0.19}$ | $33.05^{\pm0.26}$ | $34.19^{\pm0.18}$ | $27.88^{\pm0.30}$ |
| FedDyn | $80.60^{\pm0.17}$ | $77.53^{\pm0.21}$ | $81.54^{\pm0.22}$ | $79.39^{\pm0.24}$ | $48.40^{\pm0.20}$ | $45.04^{\pm0.31}$ | $46.87^{\pm0.24}$ | $43.04^{\pm0.29}$ |
| FedDC | $81.83^{\pm0.17}$ | $78.87^{\pm0.21}$ | $82.44^{\pm0.17}$ | $80.93^{\pm0.19}$ | $48.74^{\pm0.19}$ | $45.11^{\pm0.26}$ | $45.94^{\pm0.22}$ | $43.94^{\pm0.27}$ |
| FedRCL | $76.06^{\pm0.15}$ | $66.88^{\pm0.19}$ | $76.51^{\pm0.19}$ | $72.28^{\pm0.23}$ | $42.05^{\pm0.27}$ | $38.60^{\pm0.20}$ | $40.56^{\pm0.24}$ | $37.28^{\pm0.26}$ |
| FedSAM | $79.74^{\pm0.18}$ | $74.69^{\pm0.19}$ | $79.87^{\pm0.18}$ | $76.90^{\pm0.23}$ | $44.78^{\pm0.25}$ | $43.50^{\pm0.24}$ | $44.14^{\pm0.29}$ | $43.36^{\pm0.25}$ |
| MoFedSAM | $76.36^{\pm0.15}$ | $65.74^{\pm0.19}$ | $76.74^{\pm0.17}$ | $70.74^{\pm0.21}$ | $41.07^{\pm0.19}$ | $34.11^{\pm0.23}$ | $35.91^{\pm0.17}$ | $28.55^{\pm0.27}$ |
| FedGAMMA | $80.89^{\pm0.17}$ | $75.34^{\pm0.19}$ | $81.73^{\pm0.16}$ | $78.74^{\pm0.19}$ | $49.78^{\pm0.25}$ | $46.31^{\pm0.27}$ | $47.91^{\pm0.26}$ | $45.26^{\pm0.33}$ |
| FedSMOO | $\mathbf{84.17}^{\pm0.19}$ | $\mathbf{80.92}^{\pm0.17}$ | $\mathbf{84.78}^{\pm0.19}$ | $\mathbf{82.79}^{\pm0.21}$ | $\mathbf{52.31}^{\pm0.24}$ | $\mathbf{49.42}^{\pm0.20}$ | $50.59^{\pm0.21}$ | $46.08^{\pm0.25}$ |
| FedSpeed | $82.76^{\pm0.19}$ | $79.95^{\pm0.19}$ | $83.36^{\pm0.18}$ | $80.72^{\pm0.22}$ | $49.93^{\pm0.23}$ | $49.04^{\pm0.24}$ | $\mathbf{50.61}^{\pm0.23}$ | $\mathbf{46.85}^{\pm0.25}$ |
| FedLESAM | $80.11^{\pm0.23}$ | $74.35^{\pm0.22}$ | $78.35^{\pm0.21}$ | $71.23^{\pm0.25}$ | $44.35^{\pm0.19}$ | $43.75^{\pm0.21}$ | $43.97^{\pm0.23}$ | $43.21^{\pm0.22}$ |
| FedLESAM-D | $83.26^{\pm0.19}$ | $79.89^{\pm0.20}$ | $83.99^{\pm0.23}$ | $81.89^{\pm0.21}$ | $49.77^{\pm0.20}$ | $45.35^{\pm0.22}$ | $50.58^{\pm0.19}$ | $46.55^{\pm0.21}$ |
| FedLESAM-S | $83.76^{\pm0.17}$ | $79.02^{\pm0.18}$ | $83.12^{\pm0.20}$ | $81.57^{\pm0.21}$ | $49.52^{\pm0.19}$ | $47.83^{\pm0.22}$ | $48.21^{\pm0.20}$ | $45.75^{\pm0.24}$ |
| FedTOGA(ours) | $\mathbf{84.91}^{\pm0.15}$ | $\mathbf{81.78}^{\pm0.17}$ | $\mathbf{84.90}^{\pm0.19}$ | $\mathbf{83.49}^{\pm0.14}$ | $\mathbf{54.90}^{\pm0.16}$ | $\mathbf{51.00}^{\pm0.15}$ | $\mathbf{53.25}^{\pm0.17}$ | $\mathbf{49.90}^{\pm0.21}$ |

Table 2: Dirichlet coefficients $u$ selected from $\{0.1, 0.6\}$, and $c$ is the Pathological coefficient, i.e., the number of active categories in each client. The CIFAR10 has 100 clients in the left part with 10% active in each round and 200 clients in the right part with 5% active in each round.(ResNet18)
**Note**: The extended table sees Tab. 8 in Appendix.

| Method | CIFAR10 | | | | | | | |
|---|---|---|---|---|---|---|---|---|
| Partition | Dirichlet | | Pathological | | Dirichlet | | Pathological | |
| Coefficient | $u=0.6$ | $u=0.1$ | $c=6$ | $c=3$ | $u=0.6$ | $u=0.1$ | $c=6$ | $c=3$ |
| FedAvg | $79.52^{\pm0.13}$ | $76.00^{\pm0.18}$ | $79.91^{\pm0.17}$ | $74.08^{\pm0.22}$ | $75.90^{\pm0.21}$ | $72.93^{\pm0.19}$ | $77.47^{\pm0.34}$ | $71.68^{\pm0.34}$ |
| FedAdam | $77.08^{\pm0.31}$ | $73.41^{\pm0.33}$ | $77.05^{\pm0.26}$ | $72.44^{\pm0.20}$ | $75.55^{\pm0.38}$ | $69.70^{\pm0.32}$ | $75.74^{\pm0.22}$ | $70.49^{\pm0.26}$ |
| SCAFFOLD | $81.81^{\pm0.17}$ | $78.57^{\pm0.14}$ | $83.07^{\pm0.10}$ | $77.02^{\pm0.18}$ | $79.00^{\pm0.26}$ | $76.15^{\pm0.15}$ | $80.69^{\pm0.21}$ | $74.05^{\pm0.31}$ |
| FedCM | $82.97^{\pm0.21}$ | $77.82^{\pm0.16}$ | $83.44^{\pm0.17}$ | $77.82^{\pm0.19}$ | $80.52^{\pm0.29}$ | $77.28^{\pm0.22}$ | $81.76^{\pm0.25}$ | $76.72^{\pm0.25}$ |
| FedDyn | $83.22^{\pm0.18}$ | $78.08^{\pm0.19}$ | $83.18^{\pm0.17}$ | $77.63^{\pm0.14}$ | $80.69^{\pm0.23}$ | $76.82^{\pm0.17}$ | $82.21^{\pm0.18}$ | $74.93^{\pm0.22}$ |
| FedSAM | $81.46^{\pm0.12}$ | $77.03^{\pm0.17}$ | $81.13^{\pm0.23}$ | $78.30^{\pm0.24}$ | $78.32^{\pm0.16}$ | $74.00^{\pm0.14}$ | $78.75^{\pm0.27}$ | $75.12^{\pm0.29}$ |
| MoFedSAM | $85.29^{\pm0.13}$ | $80.25^{\pm0.17}$ | $84.74^{\pm0.16}$ | $83.09^{\pm0.24}$ | $84.76^{\pm0.20}$ | $\mathbf{80.10}^{\pm0.14}$ | $85.00^{\pm0.27}$ | $\mathbf{82.13}^{\pm0.23}$ |
| FedGAMMA | $82.82^{\pm0.16}$ | $79.91^{\pm0.15}$ | $83.51^{\pm0.18}$ | $77.11^{\pm0.14}$ | $80.72^{\pm0.19}$ | $76.70^{\pm0.14}$ | $81.81^{\pm0.27}$ | $77.44^{\pm0.29}$ |
| FedSMOO | $\mathbf{86.08}^{\pm0.14}$ | $\mathbf{81.80}^{\pm0.18}$ | $\mathbf{86.38}^{\pm0.15}$ | $\mathbf{82.79}^{\pm0.16}$ | $\mathbf{84.96}^{\pm0.19}$ | $79.76^{\pm0.19}$ | $84.82^{\pm0.18}$ | $81.01^{\pm0.19}$ |
| FedSpeed | $86.01^{\pm0.16}$ | $81.02^{\pm0.16}$ | $86.09^{\pm0.19}$ | $82.50^{\pm0.16}$ | $84.12^{\pm0.28}$ | $76.74^{\pm0.14}$ | $84.78^{\pm0.24}$ | $79.09^{\pm0.29}$ |
| FedLESAM | $81.04^{\pm0.19}$ | $76.92^{\pm0.16}$ | $81.37^{\pm0.17}$ | $78.21^{\pm0.21}$ | $77.80^{\pm0.18}$ | $73.73^{\pm0.22}$ | $78.44^{\pm0.20}$ | $74.53^{\pm0.19}$ |
| FedLESAM-D | $84.27^{\pm0.14}$ | $80.08^{\pm0.19}$ | $85.62^{\pm0.18}$ | $83.00^{\pm0.22}$ | $82.53^{\pm0.19}$ | $79.56^{\pm0.27}$ | $\mathbf{85.04}^{\pm0.21}$ | $81.10^{\pm0.19}$ |
| FedLESAM-S | $84.94^{\pm0.12}$ | $79.52^{\pm0.17}$ | $85.88^{\pm0.19}$ | $82.18^{\pm0.15}$ | $83.22^{\pm0.22}$ | $78.69^{\pm0.17}$ | $85.02^{\pm0.24}$ | $80.57^{\pm0.17}$ |
| FedTOGA(ours) | $\mathbf{86.99}^{\pm0.13}$ | $\mathbf{83.16}^{\pm0.17}$ | $\mathbf{87.21}^{\pm0.18}$ | $\mathbf{84.55}^{\pm0.15}$ | $\mathbf{85.21}^{\pm0.17}$ | $\mathbf{81.60}^{\pm0.16}$ | $\mathbf{85.24}^{\pm0.19}$ | $\mathbf{83.25}^{\pm0.20}$ |

Table 3: WALL-CLOCK Time Comparison. **Note**: The extended table sees Tab. 9 in Appendix.

| Method | R(80%) | Cost | R(82%) | Cost |
|---|---|---|---|---|
| FedSAM | 481 | 3.6× | 800+ | 4.7× |
| MoFedSAM | 167 | 1.2× | 270 | 1.6× |
| FedGAMMA | 458 | 3.4× | 630 | 3.7× |
| FedSpeed | 262 | 1.9× | 318 | 1.9× |
| FedSMOO | 190 | 1.4× | 253 | 1.5× |
| FedLESAM-D | 248 | 1.8× | 418 | 2.5× |
| FedTOGA | 135 | 1.0× | 170 | 1.0× |

## 6.2 PERFORMANCE EVALUATION

**Performance with compared benchmarks.** As shown in Tables 1 and 2, the proposed FedTOGA algorithm performs excellently on various heterogeneous datasets regarding convergence speed and final achieved accuracy. Table 1, which details the test accuracy of the LeNet model, demonstrates that FedTOGA significantly outperforms other algorithms with different heterogeneous data conditions. Specifically, under the Dirichlet-0.1 setting on the CIFAR10 dataset, FedTOGA attains an accuracy of 82.05%, marking a significant improvement of over 7.37% compared to vanilla FedAvg and a 1.99% increase over the second-highest baseline accuracy. Similar results are observed in Table 2 for the ResNet18 model, FedTOGA outperforms all current baseline algorithms. As seen in Table 3, FedTOGA also exhibits a significant advantage in terms of convergence speed. When reaching 80% accuracy, FedTOGA converges 3.6× faster than FedSAM and 1.2× faster than the second-best algorithm. Similarly, when reaching 82% accuracy, FedTOGA converges 4.7× faster compared to FedSAM and 1.5× faster than the second-best algorithm. This indicates that FedTOGA achieves the target accuracy with significantly reduced computation and communication overhead compared to other methods.

**Impact of heterogeneity.** We use the Dirichlet and Pathological methods for data partitioning. For the Dirichlet distribution, we adopt with variance coefficients $u$ of 0.1 and 0.6. We use coefficients $c$ of 3 and 6 for the Pathological distribution. As shown in Tables 1 and 2, increased data heterogeneity leads to decreased accuracy across all algorithms. However, FedTOGA exhibits the smallest accuracy drop. Specifically, for the Resnet18 model, as $u$ changes from 0.6 to 0.1 under the Dirichlet distribution on the CIFAR10 dataset, FedSAM's accuracy decreases from 81.46% to 77.03%, a 4.43% reduction. At the same time, the second-best algorithm, FedSMOO, shows a drop from 86.08% to 81.80%, a 4.28% reduction. In contrast, FedTOGA's accuracy declines from 86.99% to 83.16%, a 3.83% drop. Similar trends are observed under the Pathological distribution, underscoring FedTOGA's superior stability and accuracy across varying levels of data heterogeneity.

**Impact of partial participation.** We fix all hyperparameters except the client participation rate to assess its effect on accuracy. As illustrated in Table 2, a reduction in the client participation rate from 10% to 5% results in a modest decline in accuracy across all algorithms. For instance, on the CIFAR10 dataset, under the challenging pathological distribution with $c = 3$, FedTOGA's accuracy decreases marginally from 84.55% to 83.35%, a reduction of just 1.40%, while FedSMOO experiences a sharper decline from 82.79% to 81.01%, a reduction of 1.78%. Similarly, under the Dirichlet distribution with $u = 0.1$, FedTOGA's accuracy decreases from 86.99% to 85.21%, a decrease of 1.78%, whereas FedSMOO's accuracy drops from 86.08% to 84.96%, a reduction of 1.12%. Despite these reductions, FedTOGA consistently outperforms other algorithms' accuracy, highlighting its robust generalization capability and stability.

## 7 CONCLUSION

In this paper, we propose a novel FL algorithm, FedTOGA, which, for the first time, estimates global perturbations by combining global training gradients and enhances the local dynamic regularizer. This ensures local clients can effectively align with the global generalization and optimization objectives. FedTOGA facilitates the efficient search for globally consistent flat minima and accelerates convergence without incurring additional local storage or uplink communication overhead. Theoretical analysis guarantees that FedTOGA achieves a fast convergence rate of $O(1/T)$. Extensive experiments were conducted to verify its efficiency and remarkable performance.

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

## A    RELATED WORKS

### A.1    LITERATURE REVIEW

In this section, we review the contributions of related works.

**Federated Learning**  FL gained widespread attention upon its introduction due to its data-exchange-free nature. FedAvg McMahan et al. (2017) As its foundational framework, it allows for collaborative modeling without exchanging data Stich (2019). However, due to various irresistible factors, the data of cooperative devices show heterogeneous distribution, which makes the modeling effectiveness suffer. Therefore, many studies based on empirical loss minimization have been proposed to solve this problem. FedProx Li et al. (2020a) employs a simple and intuitive practice, ensuring that the local model is not far from the global model. Specifically, regular terms are introduced during local training to limit the distance between the local and global models. SCAFFOLD Karimireddy et al. (2020), Mime Karimireddy et al. (2021) uses control variables for local updates. However, it requires greater communication overhead. FedDyn Acar et al. (2021), FedPD Zhang et al. (2020) considers the inconsistency of the local optimal point with the global optimal point to be a fundamental dilemma, which aligns the locally optimal solution to the global optimal solution via a dynamic regularizer. FedPA Al-Shedivat et al. (2021) removes bias from client updates by estimating a global posterior. FedDC Gao et al. (2022) takes decoupled local and global updates to mitigate heterogeneity. Furthermore, recent research has shown that local model bias is similar to catastrophic forgetting in continuous learning Lee et al. (2022; 2024); Shoham et al. (2019); Wang et al. (2023), Clients overriding previously important parameters to learn a new task resulted in disrupting pretask performance. Some studies have mitigated global knowledge collapse by task recall Rebuffi et al. (2017); Dong et al. (2022). Server momentum-based Sun et al. (2023a) algorithms also play an important role in FL. Zaheer et al. (2018) investigates the convergence failure of ADAM in certain non-convex settings and develops an adaptive optimizer, YOGI, which aims to improve convergence. Reddi et al. (2021) integrates it in a FL framework. FedAvgM Hsu et al. (2019) using Momentum Qian (1999), while FedACG Kim et al. (2024) utilizes NAG Nesterov (1983). And FedCM Xu et al. (2021) mitigates local heterogeneity by using the proximity global update gradient applied to the client momentum. FedLADA Sun et al. (2023d) combines local ADAM with FedCM to dynamically modify local deviations.

**Sharpness-aware Minimization**.  Many studies Hochreiter & Schmidhuber (1994); Dinh et al. (2017) have pointed out that flat minima imply superior generalization performance, which possesses greater robustness to model perturbations. In order to minimize sharpness Keskar et al. (2017), Foret et al. (2021); Becker et al. (2024) proposed sharpness-aware minimization (SAM), and many works Li & Giannakis (2023); Mueller et al. (2023) were carried out. Specifically, SAM only captures the sharpness of specific small batches of data, and VaSSO Mueller et al. (2023) aims to address this issue. Our FedTOGA can also help solve this problem to some extent. First, we add neighborhood perturbations $\tilde{g}_{i,k-1}^t$ to help the local SAM optimizer perceive the amount of neighboring batch data perturbations (optionally), and global update perturbations via $\Delta^{t-1}$. In addition, m-SAM Andriushchenko & Flammarion (2022) can be considered to be closely related to federated sharpness minimization. m-SAM Andriushchenko & Flammarion (2022) quantifies the sharpness of batches across m training point batches, averaging out multiple disjoint batches in the generated Updates. The neighborhood global perturbation proposed by our FedTOGA alleviates the problem of local perturbation isolation in the FL paradigm and can be applied to all existing algorithms.

**SAM in Federated Learning**  To extend the generalizability of local models in FL, Qu et al. (2022); Caldarola et al. (2022) introduced SAM into the FL paradigm to propose FedSAM. further, Qu et al. (2022) combined with FedCM Xu et al. (2021) to propose MoFedSAM. FedGAMMA Dai et al. (2023a) introduces the variance reduction technique of SCAFFOLD Karimireddy et al. (2020) into FedSAM and gets some results. And FedSpeed Sun et al. (2023c) uses SAM to optimize FedDyn Acar et al. (2021). FedSMOO Sun et al. (2023b) builds on FedSpeed to use dynamic regularization to SAM to estimate global disturbances. FedSOL Lee et al. (2024) uses the orthogonal idea of continuous learning to make local perturbations close to the global. Fan et al. (2024a) proposes an efficient algorithm, the Local Estimation of Global Perturbations SAM (FedLESAM), which optimizes global sharpness and reduces computation. As we have seen, FedSAM Qu et al. (2022); Caldarola et al. (2022), MoFedSAM, and FedGAMMA Dai et al. (2023a) compute local perturbations and optimize sharpness on client data, which may result in the local SAM does not reach the

global flat minimum. Several studies have identified this drawback and attempted to address it. Fed-SOL Lee et al. (2024) uses local orthogonal solving to limit the range of local perturbations, which can lead to perturbation absences. FedSMOO Sun et al. (2023b) uses dynamic regularity to compute and add corrections; however, it requires additional communication and storage overheads. FedLE-SAM Fan et al. (2024a) believes that additional computation would be burdensome and, therefore, uses historical storage parameters to estimate global perturbations. However, the above solutions may have limitations due to network fluctuations, which we discuss in detail in A.2. Therefore, we propose FedTOGA to estimate the global perturbation using the global update.

Table 4: Baic Notations

| | |
|---|---|
| $i, k, t$ | Number of the client, local training interval and global epoch. |
| $\eta_l, \rho$ | Local learing rate, and perturbation learning rate. |
| $D, D_i$ | Data distributions of global and $i$-th client. |
| $h, h_i$ | Global and local dual variables. |
| $\Delta^t$ | Global update gradient in $t$-th round. |
| $\theta, \theta^t, \theta_{i,k}^t$ | Model weights and weights of global and local models. |
| $\delta$ | Perturbation towards to the sharpest point near the neighborhood of $\theta$ |

## A.2 EXISTING LIMITATIONS

**FedSMOO** Sun et al. (2023b) In the algorithm 2, to solve the model perturbation problem, Sun et al. (2023b) utilizes the local Augmented Lagrangian function to penalize the deviation of the local perturbation from the global perturbation. As training proceeds, the local perturbation is made to approach the global perturbation gradually. However, it needs to open extra storage space on the client side to record $\mu_i, \tilde{s}_i$. Meanwhile, $\tilde{s}_i$ needs to be synchronously uploaded to update the global perturbation variable $s$ at the time of aggregation during the communication process, which doubles the communication overhead. Further, this estimation bias will be exacerbated by the server's strategy of randomly selecting the set of clients $S_t$ to mitigate the communication overhead due to communication bottlenecks.

---

**Algorithm 2:** FedSMOO Algorithm

---

1 Initial $\theta^0, \theta_i, s_i, s, \lambda_i, \lambda, \mu_i$
2 **for** *each round $t \in [T] \triangleq \{0, 1, 2, \cdots, T-1\}$* **do**
3   Sample the active client set $S_t \subseteq [N]$.
4   **for** $i \in S_t$ *in parallel* **do**
5     $\theta_i^t, \tilde{s}_i \leftarrow$ **Client Update**$(\theta^t, s^t)$;    **communicate** $\theta_i^t, \tilde{s}_i$ to server ;
6   **end**
7   $\mathcal{S}^t = \frac{1}{M} \sum_{i \in S_t} \tilde{s}_i; s^t = \rho \frac{\mathcal{S}^t}{\|\mathcal{S}^t\|};$
     $h^{t+1} = h^t - \frac{1}{\alpha N} \sum_{i \in S_t} (\theta_i^t - \theta^t);$    $\theta^{t+1} = \frac{1}{M} \sum_{i \in S_t} \theta_i^t - \alpha h^{t+1};$
8 **end**
9 **Client Update**$(\theta_t, s^t)$: $\theta_{i,0}^t = \theta^t; s = s^t$
10 **for** *local epoch $k \in [K] \triangleq t\{0, 1, 2, \cdots, K-1\}$* **do**
11   sample a mini-batch data $\xi_{i,k}^t$; gradient estimate: $g_{i,k}^t = \nabla f_i(\theta_{i,k}^t; \xi_{i,k}^t)$;
12   Perturbation: $\mathcal{S}_{i,k}^t = g_{i,k}^t - \mu_i - s; \hat{s}_{i,k}^t = \rho \frac{\mathcal{S}_{i,k}^t}{\|\mathcal{S}_{i,k}^t\|}; \mu_i = \mu_i + (\hat{s}_{i,k}^t - s)$
13   extra-step: $\tilde{g}_{i,k}^t = \nabla f_i(\theta_{i,k}^t + \hat{s}_{i,k}^t; \xi_{i,k}^t);$    $\theta_{i,k+1}^t = \theta_{i,k}^t - \eta_l \left(\tilde{g}_{i,k}^t - h_i^t + \frac{1}{\alpha}\left(\theta_{i,k}^t - \theta_{i,0}^t\right)\right)$
14 **end**
15 $\tilde{s}_i = \mu_i - \hat{s}_{i,K}^t; h_i^{t+1} = h_i^t - \frac{1}{\alpha}\left(\theta_{i,K}^t - \theta_{i,0}^t\right)$
16 **return** $\theta_i^t = \theta_{i,K}^t; \tilde{s}_i$

---

**FedLESAM** Fan et al. (2024a) In the algorithm 3, to reduce the computational overhead and estimate the global perturbations, Fan et al. (2024a) utilizes the historical global model record values $\theta_i^{old}$ to compare with the latest round's global model $\theta^t$ to estimate the global perturbations. This poses the same problem as the algorithm 2 described above, specifically, in the face of an extreme case where participants will only participate in one global aggregation, FedLESAM will not be able to estimate the global perturbation variables efficiently. Meanwhile, the perturbation scales will vary when the frequency of client participation is different. In addition, since the perturbation estimation does not include the current perturbation computation, it may not be possible to accurately estimate the current perturbation direction.

---

**Algorithm 3:** FedLESAM-D Algorithm

---

1 Initial $\theta^0, \theta_i^{old}, h_i, h$
2 **for** *each round $t \in [T] \triangleq \{0, 1, 2, \cdots, T-1\}$* **do**
3   Sample the active client set $S_t \subseteq [N]$.
4   **for** $i \in S_t$ *in parallel* **do**
5     $\theta_i^t \leftarrow$ **Client Update**$(\theta^t)$;    **communicate** $\theta_i^t$ to server ;
6   **end**
7   $h^{t+1} = h^t - \frac{1}{\alpha N} \sum_{i \in S_t}(\theta_i^t - \theta^t);$    $\theta^{t+1} = \frac{1}{M} \sum_{i \in S_t} \theta_i^t - \alpha h^{t+1};$
8 **end**
9 **Client Update**$(\theta_t)$: $\theta_{i,0}^t = \theta^t$
10 **for** *local epoch $k \in [K] \triangleq \{0, 1, 2, \cdots, K-1\}$* **do**
11   sample a mini-batch data $\xi_{i,k}^t$; Perturbation: $\delta_{i,k}^t = \rho \frac{\theta_i^{old} - \theta^t}{\|\theta_i^{old} - \theta^t\|}$
12   extra-step: $\tilde{g}_{i,k}^t = \nabla f_i(\theta_{i,k}^t + \delta_{i,k}^t; \xi_{i,k}^t);$    $\theta_{i,k+1}^t = \theta_{i,k}^t - \eta_l \left(\tilde{g}_{i,k}^t - h_i^t + \frac{1}{\alpha}\left(\theta_{i,k}^t - \theta_{i,0}^t\right)\right)$
13 **end**
14 $h_i^{t+1} = h_i^t - \frac{1}{\alpha}\left(\theta_{i,K}^t - \theta_{i,0}^t\right);$    $\theta_i^{old} = \theta^t$
15 **return** $\theta_i^t = \theta_{i,K}^t$

---

Table 5: Abstract for the SAM-based FL algorithms for solving data heterogeneity, focusing on the basic algorithm, sharpness minimization objective, perturbation computation strategy, additional communication, and storage overhead comparison.

| Works | Base Algorithm | Minimizing Target | Local Perturbation | Extra-S | Extra-C |
|---|---|---|---|---|---|
| FedSAM | FedAvg | Local Sharpness | $\rho \frac{g_{i,k}^t}{\|g_{i,k}^t\|}$ | $1\times$ | $1\times$ |
| MoFedSAM | FedCM | Local Sharpness | $\rho \frac{g_{i,k}^t}{\|g_{i,k}^t\|}$ | $1\times$ | $1\times$ |
| FedSpeed | FedDyn | Local Sharpness | $\rho \frac{g_{i,k}^t}{\|g_{i,k}^t\|}$ | $2\times$ | $1\times$ |
| FedGAMMA | SCAFFOLD | Local Sharpness | $\rho \frac{g_{i,k}^t}{\|g_{i,k}^t\|}$ | $2\times$ | $2\times$ |
| FedSMOO | FedDyn | Local Sharpness With Correction | $\rho \frac{g_{i,k}^t - \mu_i - s}{\|g_{i,k}^t - \mu_i - s\|}$ | $3\times$ | $2\times$ |
| FedLESAM(S-D) | FedDyn SCAFFOLD | Global Sharpness | $\rho \frac{\theta_i^{old} - \theta_t}{\|\theta_i^{old} - \theta_t\|}$ | $3\times$ | $1\times$ |
| **FedTOGA(ours)** | FedDyn FedCM | Local With Global Sharpness Estimate | $\rho \frac{g_{i,k}^t + \kappa \Delta^t}{\|g_{i,k}^t + \kappa \Delta^t\|}$ | $2\times$ | $1\times$ |

## A.3 NEIGHBOURHOOD PERTURBATION STRATEGY

**How Neighbourhood Perturbation works?** Recall the fact that when the local iteration interval is greater than 1, the gradient register needs to be cleared by (optimizer.zero_grad()) each time the gradient computation is performed. This is to prevent the accumulating gradient from causing errors. Recognizing that using the neighborhood gradient variables in the registers does not need substantial additional overhead, we use the neighboring gradients temporarily stored in the registers to simulate the current perturbation gradient. We give an example of a local gradient computation to help better understand the workflow. We initialize the model $\theta_0$ and a gradient register $G = [\emptyset]$. After the first calculation of the gradient (loss.backward()), the register is updated $G = [g_1 = \nabla f(\theta_0)]$. If SGD is used, $\theta_1 = \theta_0 - \eta g_1$, followed by clearing the register $G = [\emptyset]$ before computing the second gradient. If SAM is used, then the perturbation is computed as $\delta_1 = \rho \frac{g_1}{\|g_1\|}$, then the gradient register is emptied, and after the gradient is computed again as $\tilde{g}_1 = \nabla f(\theta + \delta_1)$, perform model update $\theta_1 = \theta_0 - \eta \tilde{g}_1$. So the register status changes to $G = [\emptyset] \to [g_1] \to [\emptyset] \to [\tilde{g}_1]$. When neighborhood perturbation is enabled, gradient calculation and clearing before SAM are no longer required. Therefore, the register status changes to $G = [g_0] \to [\emptyset] \to [g_1]$. The client will directly use the previously calculated gradient of the gradient register as the perturbation variable. You can observe the gradient calculation and cache change process in Fig. 2.

**How does Perturbation Fusion work?** With the above technical means of neighborhood perturbation, we can easily merge it in the perturbation computation by not emptying the gradient cache before computing the perturbation, then the gradient cache state will change to $G = [\tilde{g}_0] \to [\tilde{g}_0 + g_1] \to [\emptyset] \to [\tilde{g}_1]$.

**What is LOOKAHEAD?** LOOKAHEAD Zhang et al. (2019) uses a fast-slow-step mechanism, where a retrospective is performed every $K$ steps forward. The idea behind this is to take a step in the direction of the current gradient update and then use a set of additional weights (called "slow weights") to take a step in the same direction but on a longer time scale. These slow weights are updated less frequently than the original weights, effectively creating a "look ahead" into the future of the optimization process. Incorporating N-perturbation techniques forms a Lookahead-like updating mechanism that helps the optimizer escape local minima and saddle points more efficiently, leading to faster convergence. Experiments on the fusion of N-P with existing algorithms in Sec.B.7.

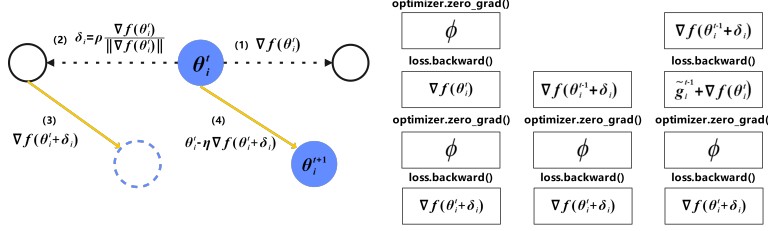

Figure 2: Illumination of the perturbation technique and its variants

## B EXPERIMENTS

### B.1 INTRODUCTION OF DATASETS

Table 6: Summary of CIFAR10/100

| Dataset | Total Number | Train Data | Test Data | Class | Size |
|---------|--------------|------------|-----------|-------|------|
| CIFAR10 | 60,000 | 50,000 | 10,000 | 10 | $3\times32\times32$ |
| CIFAR100 | 60,000 | 50,000 | 10,000 | 100 | $3\times32\times32$ |

CIFAR10 and CIFAR100 are two datasets widely used in machine learning research. As shown in Table 6, CIFAR10 contains 60,000 color images in 10 categories, with 6,000 images in each category and an image size of $32 \times 32$ pixels. CIFAR100 is similar to CIFAR10, but it contains 100 categories with 600 images per category, 500 of which are used for training and 100 for testing. These categories are further categorized into 20 major categories, each containing five subcategories.

### B.2 DETAILED HYPERPARAMETERS SELECTION

To ensure a fair comparison across different datasets, we employed an experimental design consistent with FedGAMMA Dai et al. (2023a), FedSMOO Sun et al. (2023b), and FedLESAM Fan et al. (2024a). ResNet18 He et al. (2016) was selected as the backbone model, utilizing group normalization Wu & He (2018) and stochastic gradient descent (SGD). 800 training rounds were conducted, with the initial local learning rate set to $\eta_l = 0.1$. The global learning rate was maintained at $\eta_g = 1.0$ for most experiments, except for FedAdam and FedYOGI Reddi et al. (2021) were adjusted to $0.01$. The penalty coefficients $\alpha$ for FedDC Gao et al. (2022) and FedDyn Acar et al. (2021) were uniformly set to $0.01$ in the LeNet, consistent with Gao et al. (2022), but were increased to $0.1$ for ResNet18. In FedACG Kim et al. (2024), following its prescribed settings, the local penalty coefficient was set to $\mu = 0.01$, and the server momentum coefficient $\lambda$ was set to $0.85$. For FedAdam and FedYOGI Reddi et al. (2021), the parameters were set as $\beta_1 = 0.9$, $\beta_2 = 0.99$, and $\tau = 1e^{-3}$. The momentum trade-off coefficient for FedCM Xu et al. (2021) was configured as $\alpha = 0.1$. In the SAM-based algorithms, the penalty coefficients for FedSpeed, FedSMOO, FedLESAM-D, and FedTOGA were uniformly set to $\alpha = 0.1$. The perturbation coefficients for FedGAMMA, FedSpeed, FedSMOO, FedLESAM(S-D), and FedTOGA were consistently set to $\rho = 0.1$ for ResNet18, except for FedSAM and MoFedSAM Qu et al. (2022) and the vanilla FedLESAM coefficient were set to $0.01$. In the LeNet experiments, the perturbation coefficients $\rho$ for FedGAMMA, FedLESAM, and its variants were set to $0.01$, though in some cases, $0.1$ yielded better performance. Weight decay was uniformly set to $1e^{-3}$ across all experiments. In the ResNet18 experiments, the learning rate decay was set to $0.998$ for most methods, except for FedSMOO, FedLESAM, and its variants, which were set to $0.9995$. In the 200-client case, the learning rate decay was set to $0.9995$ (This is not always the case; in some cases, a learning rate decay of $0.998$ works better, and we kept only the best results). In most scenarios, the local perturbation correction coefficient $\kappa$ for FedTOGA was set to $1$; however, in cases of increased heterogeneity, $\kappa$ could be slightly enlarged but not beyond the local interval value $K$. The local dual variable correction coefficient $\beta$ for FedTOGA was chosen from $0$ to $1$, with $0.8$ or $0.9$ typically performing best on CIFAR10. Generally, the parameter selection range can be determined according to Table 7.

Table 7: Hyperparameters Selection.

| Options | SGD-type | Best Selection | proxy-Type | Best Selection |
|---------|----------|----------------|------------|----------------|
| Local Learning Rate | {0.01,0.1,0.5} | 0.1 | {0.01,0.1,0.5} | 0.1 |
| Global Learning Rate | {0.01,0.1,1.0} | 1.0 | {0.01,0.1,1.0} | 1.0 |
| Learning Rate Decay | {0.995,0.998,0.9995} | 0.998 | {0.997,0.998,0.9995} | 0.9995 |
| SAM Learning Rate | {0.001,0.01,0.1} | 0.01 | {0.001,0.01,0.1} | 0.1 |
| penalized coefficient $\alpha$ | {0.01,0.1,0.2} | 0.1 | {0.01,0.1,0.2} | 0.1 |
| client-level momentum $\alpha$ | {0.01,0.05,0.1} | 0.1 | - | - |
| SAM Perturbation Correction $\kappa$ | - | - | {1,2,4} | 1 |
| Dual variable Correction $\beta$ | - | - | {0.1,0.5,0.8,0.9} | 0.8 |
| Server-level momentum $\lambda$ | {0.8,0.85,0.9} | 0.85 | - | - |

**Test Experiments**: Quadro RTX 6000; Driver Version 515.76; CUDA Version 11.7

## B.3 DISTRIBUTIONS OF DIRICHLET AND PATHOLOGICAL SPLIT

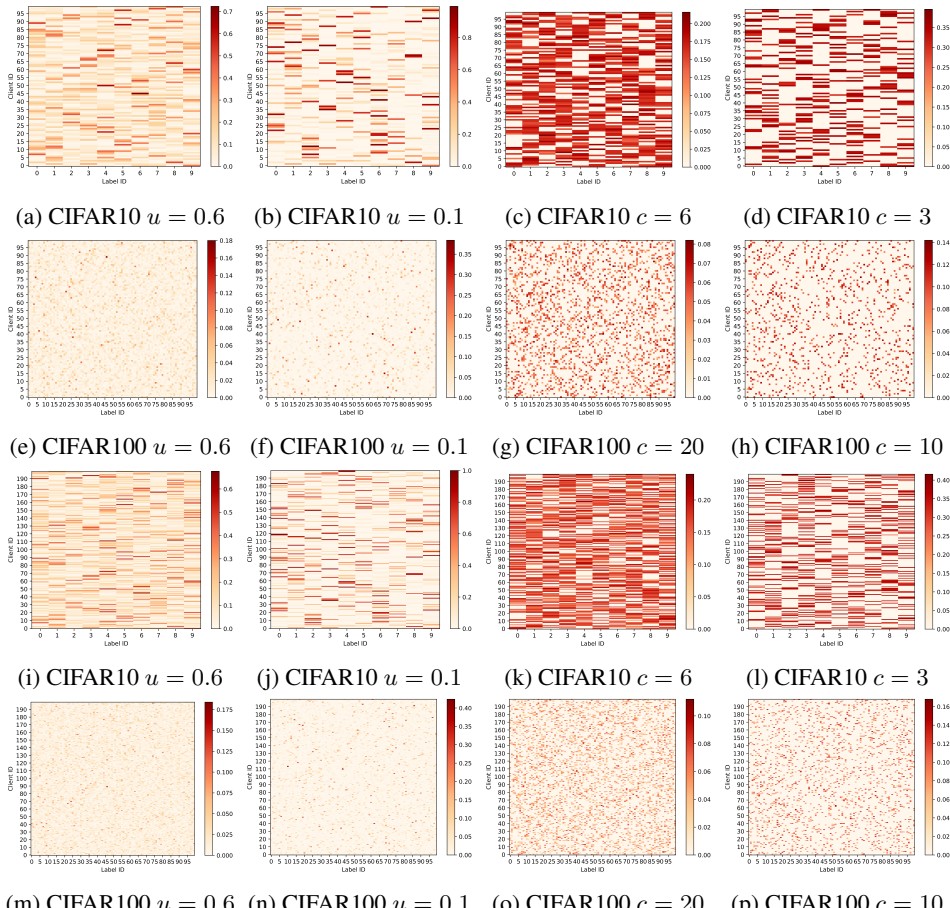

(a) CIFAR10 $u = 0.6$    (b) CIFAR10 $u = 0.1$    (c) CIFAR10 $c = 6$    (d) CIFAR10 $c = 3$

(e) CIFAR100 $u = 0.6$    (f) CIFAR100 $u = 0.1$    (g) CIFAR100 $c = 20$    (h) CIFAR100 $c = 10$

(i) CIFAR10 $u = 0.6$    (j) CIFAR10 $u = 0.1$    (k) CIFAR10 $c = 6$    (l) CIFAR10 $c = 3$

(m) CIFAR100 $u = 0.6$   (n) CIFAR100 $u = 0.1$   (o) CIFAR100 $c = 20$   (p) CIFAR100 $c = 10$

Figure 3: Heatmaps of the data distributions for ClAR10 and ClFAR100 for Dirichlet distributions with coefficients of 0.6 and 0.1, respectively, and for Pathological sampling probabilities with coefficients of 6/20 and 3/10. Both datasets consistently include 100 / 200 clients.

**Dirichlet Sampling**: The Dirichlet distribution can be thought of as the conjugate prior of a polynomial distribution, and is used to generate weights for a mixture model or to distribute samples in the context of a non-uniform category distribution. By adjusting the parameter $u$, it is possible to generate data ranging from extremely inhomogeneous (near-discrete concentration in a category) to uniformly distributed. The data exhibit a long-tailed distribution. See Fig. 3.

**Pathological Sampling**: A typical feature of pathological sampling is extreme skewness or anomalies in the data distribution, which may lead to unstable training, convergence difficulties, or severe model performance degradation. We used it to test and validate the model's performance under adverse conditions. The data are presented in species isolation; see Fig.3.

The splitting strategy for all data is consistent with FedGAMMA Dai et al. (2023a), FedSMOO Sun et al. (2023b), and FedLESAM Fan et al. (2024a).

B.4 EVALUATION CURVES

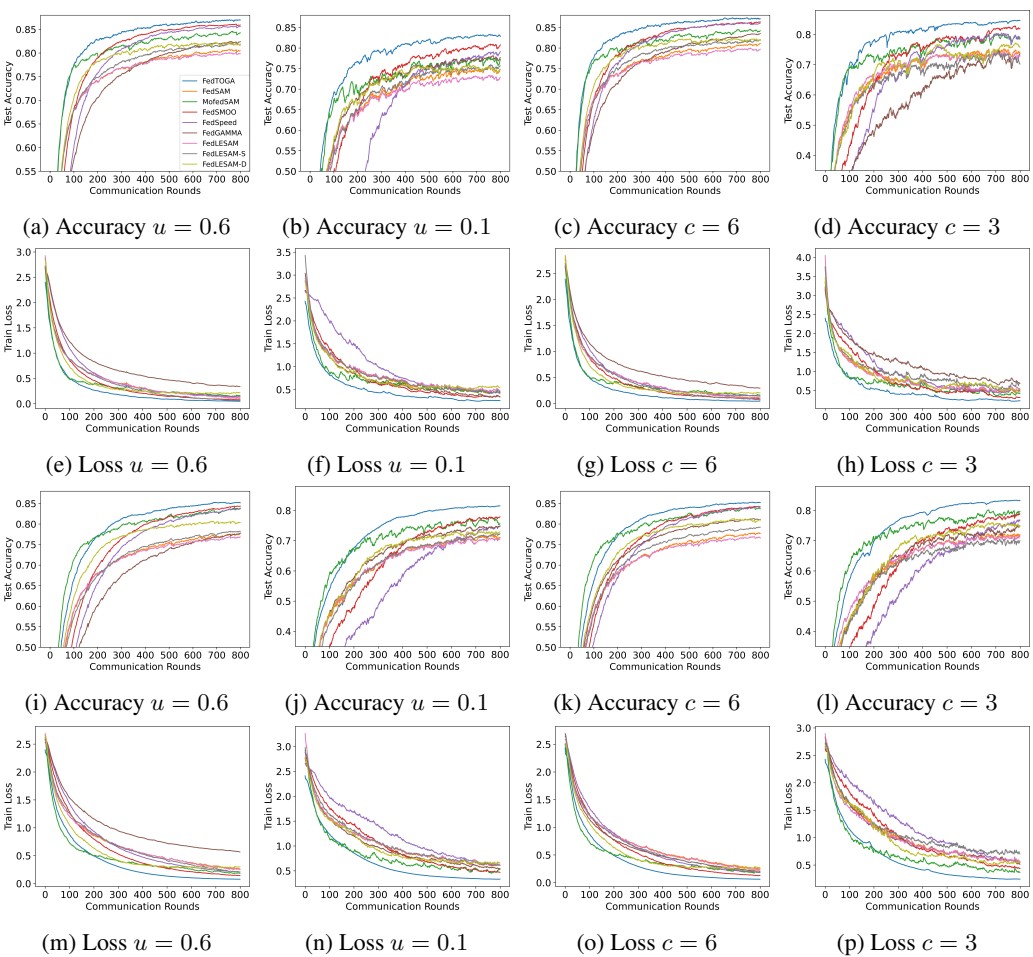

Figure 4: Accyracy/ Loss on the CIFAR-10 dataset under 10% /5% participation of total 100/200 clients

As can be seen from the above figures, FedTOGA significantly outperforms other algorithms in scenarios with significant heterogeneity (e.g., Dirichlet-0.1 and Pathological-3). Our algorithm still shows stability even when the number of clients decreases(e.g., 200 clients with 5% participation). These results are in line with our expectations. We aim to design an algorithm that enhances global consistency while efficiently finding a global flat minimum to improve generalization and reduce edge node computation and storage requirements.

In addition to the results in the main text, we also conducted experiments on CIFAR100. We followed the same parameter settings with FedSMOO, but we found that the baseline produced fluctuations in performance. Therefore, we report the best performance of previous studies (OLD) and the results of brand new experiments (NEW) in table 8. To show the extraordinary performance of FedTOGA, we only report the historical best accuracy of all benchmarks in the main text.

Table 8: Test accuracy on CIFAR10/100 after 800 rounds under Dirichlet distribution and Pathological splits $u$ is the Dirichlet coefficient selected from $\{0.1, 0.6\}$ and $c$ is the Pathological coefficient, which is the number of active categories in each client. The two datasets are divided into 100 clients, and 10% of them are active at each round in the upper part, while 200 and 5% in the lower part.(ResNet18)

**CIFAR10**

| Method | Dirichlet $u=0.6$ OLD | NEW | $u=0.1$ OLD | NEW | Pathological $c=6$ OLD | NEW | $c=3$ OLD | NEW |
|---|---|---|---|---|---|---|---|---|
| FedAvg | 79.52±0.13 | 76.00±0.18 | | | 74.08±0.22 | 79.91±0.17 | | |
| FedAdam | 77.08±0.31 | 73.41±0.33 | | | 72.44±0.29 | 77.05±0.26 | | |
| SCAFFOLD | 81.81±0.17 | 78.57±0.14 | | | 77.02±0.18 | 83.07±0.10 | | |
| FedCM | 82.97±0.21 | 77.82±0.16 | | | 77.82±0.19 | 83.44±0.17 | | |
| FedDyn | 83.22±0.18 | 78.08±0.19 | | | 77.63±0.14 | 83.18±0.17 | | |
| FedSAM | 80.10±0.12 | 81.46±0.12 | 76.86±0.16 | 77.03±0.17 | 75.51±0.24 | 78.30±0.24 | 81.13±0.23 | 81.13±0.23 |
| MoFedSAM | 84.13±0.13 | 85.29±0.13 | 78.71±0.15 | 80.25±0.17 | 84.82±0.14 | 84.74±0.14 | 79.57±0.18 | 83.09±0.24 |
| FedGAMMA | 82.64±0.14 | 82.82±0.16 | 78.95±0.15 | 78.87±0.19 | 83.24±0.19 | 83.51±0.18 | 78.81±0.14 | 82.21±0.21 |
| FedSMOO | 84.55±0.14 | 86.08±0.20 | 80.82±0.17 | 81.80±0.20 | 85.39±0.21 | 86.38±0.20 | 81.58±0.16 | 82.79±0.16 |
| FedSpeed | – | 86.01±0.16 | – | 81.02±0.16 | – | 86.09±0.19 | – | 82.50±0.16 |
| FedLESAM | 81.04±0.19 | 80.94±0.16 | 76.93±0.16 | 75.93±0.16 | 81.37±0.17 | 80.92±0.19 | 77.30±0.22 | 78.21±0.21 |
| FedLESAM-D | 84.27±0.14 | 83.60±0.16 | 80.08±0.19 | 78.87±0.19 | 85.62±0.18 | 83.66±0.18 | 83.00±0.22 | 82.21±0.21 |
| FedLESAM-S | 84.94±0.12 | 83.66±0.13 | 79.52±0.17 | 78.77±0.16 | 85.88±0.19 | 82.99±0.19 | 82.18±0.15 | 81.01±0.17 |
| FedTOGA(ours) | **85.21**±0.17 | **86.99**±0.13 | **81.60**±0.16 | **83.16**±0.17 | **85.24**±0.19 | **87.21**±0.18 | **83.25**±0.19 | **84.55**±0.15 |

**CIFAR100**

| Method | Dirichlet $u=0.6$ OLD | NEW | $u=0.1$ OLD | NEW | Pathological $c=20$ OLD | NEW | $c=10$ OLD | NEW |
|---|---|---|---|---|---|---|---|---|
| FedAvg | 46.35±0.15 | 42.64±0.22 | 44.15±0.79 | | | | 40.23±0.31 | |
| FedAdam | 48.35±0.17 | 40.77±0.31 | 41.26±0.30 | | | | 32.58±0.22 | |
| SCAFFOLD | 51.98±0.23 | 44.41±0.15 | 46.06±0.22 | | | | 41.08±0.24 | |
| FedCM | 51.56±0.20 | 43.03±0.26 | 44.94±0.14 | | | | 38.35±0.27 | |
| FedDyn | 50.82±0.19 | 42.50±0.28 | 44.19±0.19 | | | | 38.68±0.14 | |
| FedSAM | 47.51±0.26 | 43.43±0.12 | 45.46±0.29 | 44.56±0.20 | 47.68±0.29 | 45.46±0.29 | 40.44±0.23 | 42.35±0.25 |
| MoFedSAM | 54.38±0.25 | 54.50±0.22 | 47.42±0.26 | 46.25±0.22 | 50.39±0.30 | 47.42±0.26 | 41.17±0.22 | 43.76±0.21 |
| FedGAMMA | 52.71±0.20 | 46.39±0.19 | 46.93±0.17 | 48.41±0.14 | 48.89±0.17 | 46.93±0.19 | 43.24±0.22 | 43.18±0.25 |
| FedSMOO | 53.92±0.18 | 46.48±0.13 | 48.87±0.17 | 47.94±0.15 | 49.01±0.20 | 48.87±0.13 | 44.10±0.19 | 43.40±0.19 |
| FedSpeed | – | 52.27±0.18 | – | 46.61±0.19 | – | 47.57±0.21 | – | 41.00±0.20 |
| FedLESAM | 47.92±0.19 | 44.48±0.20 | 44.28±0.21 | 44.28±0.20 | 46.40±0.20 | 41.20±0.18 | 41.20±0.18 | 39.65±0.19 |
| FedLESAM-D | 53.27±0.17 | 46.42±0.17 | 48.26±0.18 | 44.45±0.23 | 45.25±0.23 | 48.26±0.18 | 43.26±0.18 | 40.98±0.20 |
| FedLESAM-S | 54.61±0.20 | 48.07±0.19 | 50.26±0.18 | 47.48±0.18 | 49.88±0.19 | 50.26±0.18 | 44.42±0.17 | 43.96±0.17 |
| FedTOGA(ours) | **55.40**±0.17 | **52.91**±0.19 | **48.72**±0.17 | **44.20**±0.22 | **51.50**±0.18 | **48.84**±0.22 | **45.30**±0.22 | **43.25**±0.20 |

**CIFAR10 (200 clients, 5% active)**

| Method | Dirichlet $u=0.6$ OLD | NEW | $u=0.1$ OLD | NEW | Pathological $c=6$ OLD | NEW | $c=3$ OLD | NEW |
|---|---|---|---|---|---|---|---|---|
| FedAvg | 75.90±0.21 | 72.93±0.19 | | | 71.68±0.34 | 77.47±0.34 | | |
| FedAdam | 75.55±0.38 | 69.70±0.32 | | | 70.49±0.26 | 75.74±0.22 | | |
| SCAFFOLD | 79.00±0.26 | 76.15±0.15 | | | 74.05±0.31 | 80.69±0.21 | | |
| FedCM | 80.52±0.29 | 77.28±0.22 | | | 76.72±0.25 | 81.76±0.14 | | |
| FedDyn | 80.69±0.23 | 76.82±0.17 | | | 74.93±0.22 | 82.21±0.18 | | |
| FedSAM | 76.32±0.16 | 73.44±0.14 | 74.00±0.14 | 78.16±0.27 | 72.41±0.29 | 75.12±0.12 | 78.75±0.27 | 78.75±0.27 |
| MoFedSAM | 82.58±0.21 | 84.76±0.24 | 78.43±0.24 | 80.10±0.14 | 85.00±0.27 | 82.13±0.19 | 79.93±0.19 | 82.13±0.19 |
| FedGAMMA | 80.72±0.19 | 78.31±0.19 | 76.41±0.17 | 76.70±0.14 | 81.59±0.17 | 77.44±0.29 | 76.58±0.21 | 77.44±0.29 |
| FedSMOO | 82.94±0.19 | 84.96±0.19 | 79.76±0.19 | 77.90±0.14 | 84.82±0.18 | 78.91±0.29 | 81.01±0.19 | 78.91±0.29 |
| FedSpeed | – | 84.12±0.18 | – | 76.74±0.14 | – | 84.78±0.27 | – | 79.09±0.29 |
| FedLESAM | 77.74±0.18 | 73.73±0.22 | 73.73±0.22 | 73.03±0.17 | 84.44±0.20 | 74.47±0.29 | 74.53±0.19 | 74.47±0.29 |
| FedLESAM-D | 82.53±0.19 | 79.56±0.27 | 79.56±0.27 | 75.17±0.14 | 85.04±0.21 | 77.93±0.29 | 81.10±0.19 | 77.93±0.29 |
| FedLESAM-S | 83.22±0.22 | 78.69±0.18 | 78.69±0.18 | 73.80±0.14 | 85.02±0.24 | 74.62±0.29 | 80.57±0.17 | 74.62±0.29 |
| FedTOGA(ours) | **85.24**±0.19 | **85.21**±0.17 | **81.60**±0.16 | **83.16**±0.17 | **85.24**±0.19 | **87.21**±0.18 | **83.25**±0.19 | **84.55**±0.15 |

**CIFAR100 (200 clients, 5% active)**

| Method | Dirichlet $u=0.6$ OLD | NEW | $u=0.1$ OLD | NEW | Pathological $c=20$ OLD | NEW | $c=10$ OLD | NEW |
|---|---|---|---|---|---|---|---|---|
| FedAvg | 44.70±0.22 | 40.41±0.33 | 38.32±0.25 | | | | 36.79±0.32 | |
| FedAdam | 44.33±0.26 | 38.04±0.25 | 35.14±0.16 | | | | 30.28±0.28 | |
| SCAFFOLD | 50.70±0.18 | 41.83±0.29 | 39.63±0.31 | | | | 37.98±0.36 | |
| FedCM | 50.93±0.31 | 42.33±0.19 | 42.01±0.17 | | | | 38.35±0.24 | |
| FedDyn | 47.32±0.18 | 41.74±0.21 | 41.55±0.18 | | | | 38.09±0.27 | |
| FedSAM | 45.98±0.27 | 40.22±0.27 | 41.13±0.27 | 41.13±0.23 | 43.20±0.23 | 38.71±0.23 | 36.90±0.29 | 38.67±0.29 |
| MoFedSAM | 53.51±0.25 | 42.22±0.23 | 42.23±0.23 | 42.23±0.23 | 45.39±0.23 | 42.77±0.27 | 39.81±0.21 | 38.83±0.25 |
| FedGAMMA | 50.61±0.19 | 43.77±0.19 | 42.57±0.19 | 42.57±0.19 | 40.99±0.25 | 43.35±0.24 | 38.46±0.22 | 35.32±0.30 |
| FedSMOO | 53.45±0.19 | 45.83±0.18 | 44.03±0.19 | 44.03±0.19 | 43.68±0.20 | 44.70±0.21 | 43.41±0.22 | 42.20±0.22 |
| FedSpeed | – | 52.63±0.19 | – | 44.12±0.18 | – | 41.62±0.24 | – | 39.25±0.25 |
| FedLESAM | 45.00±0.16 | 41.87±0.23 | 40.99±0.22 | 40.99±0.22 | 42.14±0.18 | 42.71±0.25 | 39.32±0.24 | 38.16±0.22 |
| FedLESAM-D | 51.14±0.20 | 45.09±0.24 | 43.89±0.10 | 43.89±0.25 | 41.00±0.24 | 42.63±0.29 | 39.63±0.30 | 39.63±0.30 |
| FedLESAM-S | 52.26±0.18 | 44.82±0.20 | 45.68±0.23 | 42.22±0.23 | 44.96±0.18 | 43.89±0.23 | 42.75±0.22 | 42.75±0.22 |
| FedTOGA(ours) | **52.91**±0.19 | **52.91**±0.22 | **44.20**±0.22 | **44.20**±0.22 | **48.84**±0.22 | **48.84**±0.22 | **43.25**±0.20 | **43.25**±0.20 |

**Note**: Since replication in the same experimental setting produced differences in performance, we report in the main text the best performance of all experiments in previous studies.

### B.5 TRAINING SPEED

Table 9: Number of communication rounds to achieve a target accuracy. We recorded the first round of communication to reach a target accuracy. We improved the number of training rounds compared to the other algorithms in the Dirichlet-0.1/0.6 and Pathological-6.0/3.0 settings. We mainly compared the SAM-based FL algorithms.

| Partition | Dirichlet | | | | | | | | Pathological | | | | | | | |
|---|---|---|---|---|---|---|---|---|---|---|---|---|---|---|---|---|
| Coefficient | $u = 0.6$ | | | | $u = 0.1$ | | | | $c = 6$ | | | | $c = 3$ | | | |
| Acc/Rounds | 80% | cost | 82% | cost | 76% | cost | 78% | cost | 80% | cost | 82% | cost | 76% | cost | 78% | cost |
| FedSAM | 481 | 3.6× | 800+ | 4.7× | 587 | 3.2× | 800+ | 3.5× | 443 | 3.3× | 790 | 4.8× | 465 | 2.9× | 691 | 3.5× |
| MoFedSAM | 167 | 1.2× | 270 | 1.6× | 303 | 1.6× | 425 | 2.9× | 135 | 1.0× | 253 | 1.5× | 167 | 1.1× | 265 | 1.3× |
| FedGAMMA | 458 | 3.4× | 630 | 3.7× | 369 | 2.0× | 591 | 2.6× | 407 | 3.0× | 550 | 3.3× | 701 | 4.4× | 800+ | 4.0× |
| FedSMOO | 190 | 1.4× | 253 | 1.5× | 302 | 1.6× | 402 | 1.8× | 205 | 1.5× | 263 | 1.6× | 262 | 1.7× | 322 | 1.6× |
| FedSpeed | 262 | 1.9× | 318 | 1.9× | 445 | 2.4× | 530 | 2.3× | 233 | 1.7× | 292 | 1.8× | 349 | 2.2× | 438 | 2.2× |
| FedLESAM | 588 | 4.4× | 800+ | 4.7× | 800+ | 4.3× | 800+ | 3.5× | 620 | 4.6× | 800+ | 4.8× | 497 | 3.1× | 778 | 3.9× |
| FedLESAM-D | 248 | 1.8× | 418 | 2.5× | 369 | 2.0× | 663 | 2.9× | 224 | 1.7× | 376 | 2.3× | 393 | 2.5× | 452 | 2.3× |
| FedLESAM-S | 390 | 2.9× | 643 | 3.8× | 529 | 2.8× | 800+ | 3.5× | 348 | 2.6× | 602 | 3.6× | 497 | 3.1× | 800+ | 4.0× |
| **FedTOGA** | 135 | 1.0× | 170 | 1.0× | 184 | 1.0× | 226 | 1.0× | 134 | 1.0× | 166 | 1.0× | 158 | 1.0× | 200 | 1.0× |

**Note**: The SGD method is not considered.

According to the above table 9, we can see that FedTOGA performs far better than the rest of the algorithms. It has the fastest convergence rate while maintaining high accuracy. The SAM optimizer usually slows down the whole training process due to the need to compute additional perturbations to the ascent process, which will be improved by enhancing consistency. MoFedSAM enforces consistency by employing global momentum on each local client and weighting it by a factor a (usually 0.1), which means that local knowledge will be forcibly overwritten by global gradient while speeding up convergence in the early stage. However, it may not be able to draw further adequate learning progress from the locals in the later stage. FedTOGA corrects local perturbations and dynamic regularizers by guiding global updates, greatly enhancing the consistency of generalization and optimization. Therefore, our method can effectively accelerate the modeling speed and improve the modeling accuracy, especially in the case of large-scale heterogeneous. Table 10 shows that FedTOGA has a similar local computation time as the SAM-based FL algorithm.

Table 10: wall clock time on CIFAR10 ResNet18 $u = 0.6$, 0.1 and 100 clients.

| | FedSAM | MoFedSAM | FedGAMMA | FedSpeed | FedSMOO | FedLESAM-D | **FedTOGA** |
|---|---|---|---|---|---|---|---|
| time | 25.71s | 28.73s | 29.88s | 28.98s | 29.67s | 25.70s | 29.12s |

### B.6 ABLATION STUDIES

Table 11: Ablation studies of different modules.

| SAM | Dynamic Regularization | Dual Correction | SAM Correction | CIFAR10 Acc | CIFAR100 Acc |
|---|---|---|---|---|---|
| ✓ | - | - | - | 81.39% | 48.08% |
| ✓ | ✓ | - | - | 84.14% | 53.79% |
| ✓ | ✓ | ✓ | - | 85.54% | 56.85% |
| ✓ | ✓ | ✓ | ✓ | 86.01% | 57.25% |

We tested the performance of different modules called "SAM," "Dynamic Regularization," "Dual Variable Correction," and "SAM Perturbation Correction" modules on the Dirichlet 0.6 partitioned CIFAR-10/100 dataset, LeNet network. The benchmark is FedSAM. After the sequential introduction of the different modules, the CIFAR10 accuracy increased by 2.75%, 4.15%, and 4.62% compared to the FedSAM; the CIFAR100 accuracy increased by 5.71%, 8.77%, and 9.17% compared to the FedSAM.

## B.7 NEIGHBOURHOOD PERTURBATION ANALYSIS

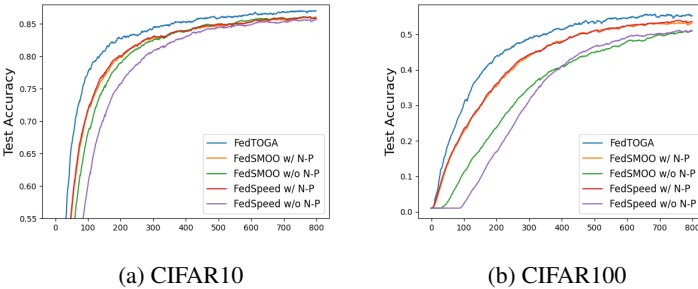

(a) CIFAR10        (b) CIFAR100

Figure 5: Testing the Impact of FedSpeed and FedSMOO with N-Perturbation Modules on CIFAR 10/100, Dirichlet 0.6, 100 Client 0.1 Participation

As shown in Fig.5. We test the performance of SAM-based FL algorithms with enabled **Neighborhood Perturbation** technology. We found that allowing neighborhood perturbation can effectively improve the performance of FedSpeedSun et al. (2023c) and FedSMOOSun et al. (2023b). This confirms our conjecture in Sec.4.2.

## B.8 HYPERPARAMETERS SENSITIVITY

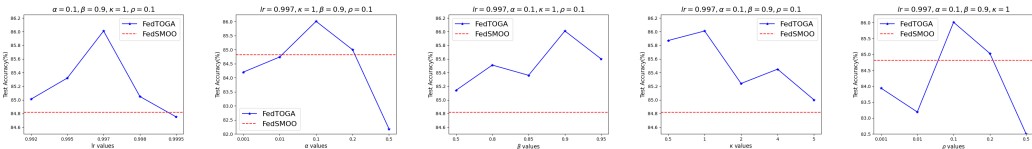

Figure 6: Hyperparameters sensitivity studies of learning rate decay, penalized coefficient $\alpha$, Correction coeficient $\beta, \kappa$ and perturbations coefficient $\rho$ on CIFAR-10.

We study the sensitivity of the hyperparameters: learning rate decay, penalty coefficient $\alpha$, correction coefficients $\beta$ and $\kappa$, and perturbation coefficient $\rho$. As shown in Figure 6, our extensive experiments demonstrate FedTOGA's resilience to variations in these hyperparameters. By systematically adjusting each parameter while holding the others constant, FedTOGA remains remarkably stable under changes in learning rate decay and the correction coefficients $\beta$ and $\kappa$. Additionally, the penalty coefficient $\alpha$ and perturbation coefficient $\rho$ effectively maintain robust performance when appropriately selected.

## B.9 DISCUSSION WITH OTHER RELATED WORKS

We show how to generalize FedTOGA to **FedDyn/ FedPD/ FedProx** without considering local perturbations, recalling the AL function defined in Eqn.(7). By setting $h_i \equiv 0; \Delta^t \equiv 0$ (i.e., omitting lines 18 and 21 in Alg.1), the local training problem of FedProx is recovered. Additionally, setting $\Delta^t \equiv 0$ recovers the local training problems of FedPD and FedDyn. When the value of $1/\alpha$ is set to zero, the local training problem of FedAvg is restored. These terms revisiting the core challenge of heterogeneous FL: local consensus inconsistency. In addition to the quadratic proximal term introduced by FedProx, FedDyn, and FedPD employ dual variables, which benefits in guiding local model updates are discussed in Sec.2.2. However, focusing solely on local stationary points is insufficient, due to the inability of clients in real FL to guarantee convergence to local stationary points after each training. Therefore, we further introduce global stationary conditions to enhance local consensus. The advantage of this approach is that clients are not burdened with additional storage or computation overhead, while the extra uplink overhead is effectively reduced, alleviating the communication bottleneck.

## B.10 LOCAL INTERVAL STUDIES

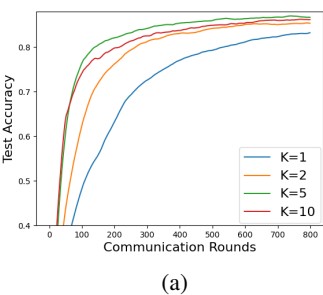 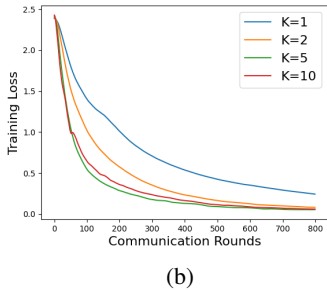

(a)                                       (b)

Figure 7: Test accuracy and training loss of FedTOGA with different intervals $K$ on CIFAR10. $K$ is set as $1, 2, 5, 10$, and other parameters are the same as mentioned above.

$K$ measures the communication interval, which refers to the number of local training steps. In Theorem 2, we observe that increasing $K$ can help the global model achieve a higher convergence rate when $T$ is large enough. However, although increasing $K$ improves the convergence speed, it also amplifies the negative effects of local heterogeneity. Figure 7 shows the impact of different values of $K$. Some previous studies suggested making $K$ large enough to approach the suboptimal value of the objective function. However, in most practical FL setups, $K$ represents a trade-off between training convergence speed and local overfitting. In our experiments, when $K = 2$, the training convergence rate and generalization of FedTOGA improved compared to $K = 1$. When $K$ increased to 5, the convergence rate was about 1.5 times faster than $K = 2$, achieving the best accuracy, which aligns with our theoretical analysis. As $K$ increases, when $K = 10$, the acceleration effect remains but becomes less significant, while generalization performance starts to decline. We believe that a larger $K$ means more local updates, which forces local clients to move toward their local optima, interfering with generalization. As the communication intervals increase, the model accuracy does not significantly decline, which demonstrates FedTOGA's robustness in long-interval communication scenarios and highlights the importance of enhancing global generalization consistency.

## B.11 MODEL DIVERGENCE

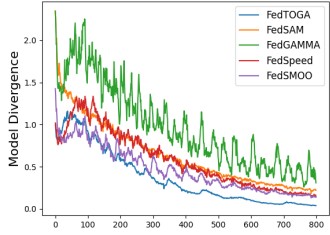

Figure 8: Consistency of Models

In the Figure 8, we show the difference between the local model and the global model after each training. Obviously, FedTOGA effectively alleviates the model's migration.

## C  PROOF FOR ANALYSIS

### C.1  THEOREM 1'S PROOF

*Proof.* Calculated by SAM rules, $\tilde{\theta}_{i,k-1}^t = \theta_{i,k-1}^t + \rho\frac{\nabla f_i(\theta_{i,k-1}^t)}{\|\nabla f_i(\theta_{i,k-1}^t)\|}$. For the induction, we assume that $\|\theta_{i,k-1}^t - v_{k-1}^t\|^2 \leq H_i(k-1)$, then

$$\|\theta_{i,k}^t - v_k^t\|^2$$

$$=\|\theta_{i,k-1}^t - \eta_l\nabla f_i(\tilde{\theta}_{i,k-1}^t) - (v_{k-1}^t - \eta_l\nabla f_i(\tilde{v}_{k-1}^t))\|^2$$

$$=\left\|\theta_{i,k-1}^t - \eta_l\nabla f_i\left(\theta_{i,k-1}^t + \rho\frac{\nabla f_i(\theta_{i,k-1}^t)}{\|\nabla f_i(\theta_{i,k-1}^t)\|}\right) - \left(v_{k-1}^t - \eta_l\nabla f\left(v_{k-1}^t + \rho\frac{\nabla f(v_{k-1}^t)}{\|\nabla f(v_{k-1}^t)\|}\right)\right)\right\|^2$$

$$=\left\|\theta_{i,k-1}^t - \eta_l\nabla f_i\left(\theta_{i,k-1}^t + \rho\frac{\nabla f_i(\theta_{i,k-1}^t)}{\|\nabla f_i(\theta_{i,k-1}^t)\|}\right) - v_{i,k-1}^t + \eta_l\nabla f\left(v_{k-1}^t + \rho\frac{\nabla f(v_{k-1}^t)}{\|\nabla f(v_{k-1}^t)\|}\right)\right\|^2$$

$$=\left\|\theta_{i,k-1}^t - v_{i,k-1}^t - \left(\eta_l\nabla f_i\left(\theta_{i,k-1}^t + \rho\frac{\nabla f_i(\theta_{i,k-1}^t)}{\|\nabla f_i(\theta_{i,k-1}^t)\|}\right) - \eta_l\nabla f_i\left(v_{k-1}^t + \rho\frac{\nabla f(v_{k-1}^t)}{\|\nabla f(v_{k-1}^t)\|}\right)\right.\right.$$

$$\left.\left.+\eta_l\nabla f_i\left(v_{k-1}^t + \rho\frac{\nabla f(v_{k-1}^t)}{\|\nabla f(v_{k-1}^t)\|}\right) - \eta_l\nabla f\left(v_{k-1}^t + \rho\frac{\nabla f(v_{k-1}^t)}{\|\nabla f(v_{k-1}^t)\|}\right)\right)\right\|^2$$

$$\leq\|\theta_{i,k-1}^t - v_{i,k-1}^t\|^2 + \eta_l^2\left\|\nabla f_i\left(\theta_{i,k-1}^t + \rho\frac{\nabla f_i(\theta_{i,k-1}^t)}{\|\nabla f_i(\theta_{i,k-1}^t)\|}\right) - \nabla f_i\left(v_{k-1}^t + \rho\frac{\nabla f(v_{k-1}^t)}{\|\nabla f(v_{k-1}^t)\|}\right)\right\|$$

$$+\eta_l^2\left\|\nabla f_i\left(v_{k-1}^t + \rho\frac{\nabla f(v_{k-1}^t)}{\|\nabla f(v_{k-1}^t)\|}\right) - \nabla f\left(v_{k-1}^t + \rho\frac{\nabla f(v_{k-1}^t)}{\|\nabla f(v_{k-1}^t)\|}\right)\right\|^2$$

$$\overset{(1)}{\leq}\|\theta_{i,k-1}^t - v_{i,k-1}^t\|^2 + \eta_l^2 L^2\left\|\theta_{i,k-1}^t - v_{i,k-1}^t + \rho\frac{\nabla f_i(\theta_{i,k-1}^t)}{\|\nabla f_i(\theta_{i,k-1}^t)\|} - \rho\frac{\nabla f(v_{k-1}^t)}{\|\nabla f(v_{k-1}^t)\|}\right\|^2 + \eta_l^2\sigma_g^2$$

$$\overset{(2)}{\leq}(1 + 2\eta_l^2 L^2)\|\theta_{i,k-1}^t - v_{i,k-1}^t\|^2 + \eta_l^2(L^2\rho^2\sigma_g'^2 + \sigma_g^2)$$

$$\leq(1 + 2\eta_l^2 L^2)\frac{L^2\rho^2\sigma_g'^2 + \sigma_g^2}{2L^2}((1 + 2\eta_l^2 L^2)^{k-1} - 1) + \eta_l^2(L^2\rho^2\sigma_g'^2 + \sigma_g^2)$$

$$\leq\frac{L^2\rho^2\sigma_g'^2 + \sigma_g^2}{2L^2}(1 + 2\eta_l^2 L^2)^k - (1 + 2\eta_l^2 L^2)\frac{L^2\rho^2\sigma_g'^2 + \sigma_g^2}{2L^2} + \eta_l^2(L^2\rho^2\sigma_g'^2 + \sigma_g^2)$$

$$=\frac{L^2\rho^2\sigma_g'^2 + \sigma_g^2}{2L^2}(1 + 2\eta_l^2 L^2)^k - \frac{L^2\rho^2\sigma_g'^2 + \sigma_g^2}{2L^2}$$

$$=\frac{L^2\rho^2\sigma_g'^2 + \sigma_g^2}{2L^2}((1 + 2\eta_l^2 L^2)^k - 1) \tag{11}$$

Equation (1-2) holds due to the Assumption 1. Recursively, it follows that the Theorem 1 holds. □

## C.2 THEOREM 2'S PROOF

Recalling the Algorithm 1, based on the FL paradigm, we propose an Augmented Lagrangian(AL) function:

$$f(\theta, h) \triangleq \frac{1}{N} \sum_{i \in N} f_i(\theta, \theta_i, h_i); F_i(\theta, \theta_i, h_i) \triangleq f_i(\theta_i) + \langle h_i, \theta - \theta_i \rangle + \frac{1}{\alpha} \|\theta_i - \theta\|^2 \quad (12)$$

By fixing $\theta$, the AL function can be separated into local pairs $\{\theta_i, h_i\}$. However, the local optimization function cannot perceive the global gradient trend due to the CTA policy. Therefore, we direct the global update $\Delta$ to be merged into the local dual variables. Thus, the local AL function is rewritten as:

$$F_i(\theta, \theta_i, h_i) \triangleq f_i(\theta_i) + \langle h_i - \beta \Delta, \theta - \theta_i \rangle + \frac{1}{\alpha} \|\theta_i - \theta\|^2 \quad (13)$$

Unlike Wang et al. (2022); Gong et al. (2022); Acar et al. (2021); Sun et al. (2023b), we relax the strict assumption that $\tilde{g}_i^t - h_i^t + \frac{1}{\alpha}(\theta_i^t - \theta^t) + \beta \Delta^t = 0$ as a strict assumption and extend the local interval to $K$ rounds, so we obtain the workflow in Algorithm 1.

First, we give all the lemmas needed for proof analysis.

**Lemma C.1.** *For $\forall \theta_{i,k}^t \in \mathbb{R}^d$ and $i$ in $S_t$, we have $\psi_{i,k}^t = \theta_{i,k}^t - \theta_{i,k-1}^t$ with the fact $\psi_{i,0}^t = 0$, and $\Psi_{i,K}^t = \sum_{k=0}^{K-1} \psi_{i,k}^t = \theta_{i,K}^t - \theta_{i,0}^t$, under the workflow in Algorithm 1, we have:*

$$\Psi_{i,K}^t = -\alpha \gamma \sum_{k=0}^{K-1} \frac{\gamma_k}{\gamma} \tilde{g}_{i,k}^t + \gamma \alpha (h_i^t - \beta \Delta^t) \quad (14)$$

*where $\gamma = \sum_{k=0}^{K-1} \gamma_k = \sum_{k=0}^{K-1} \frac{\eta_l}{\alpha} \left(1 - \frac{\eta_l}{\alpha}\right)^{K-1-k} = 1 - \left(1 - \frac{\eta_l}{\alpha}\right)^K$.*

*Proof.* According to the update rule of Line.19 in Algorithm 1, have

$$\psi_{i,k}^t = \Psi_{i,k}^t - \Psi_{i,k-1}^t = \theta_{i,k}^t - \theta_{i,k-1}^t = -\eta_l \left(\tilde{g}_{i,k}^t - h_i^t + \frac{1}{\alpha}\left(\theta_{i,k}^t - \theta_{i,0}^t\right) + \beta \Delta^t\right)$$

$$= -\eta_l \left(\tilde{g}_{i,k}^t - h_i^t + \frac{1}{\alpha}\Psi_{i,k-1}^t + \beta \Delta^t\right).$$

Then we can build the $\Psi_{i,k}^t$ as:

$$\Psi_{i,k}^t = \Psi_{i,k-1}^t - \eta_l \left(\tilde{g}_{i,k}^t - h_i^t + \frac{1}{\alpha}\Psi_{i,k-1}^t + \beta \Delta^t\right) = \left(1 - \frac{\eta_l}{\alpha}\right) \Psi_{i,k-1}^t - \eta_l(\tilde{g}_{i,k}^t - h_i^t + \beta \Delta^t).$$

Taking the iteration on $k$,

$$\theta_{i,K}^t - \theta_{i,0}^t = \Psi_{i,K}^t = \left(1 - \frac{\eta_l}{\alpha}\right)^K \Psi_{i,0}^t - \eta_l \sum_{k=0}^{K-1} \left(1 - \frac{\eta_l}{\alpha}\right)^{K-1-k} (\tilde{g}_{i,k}^t - h_i^t + \beta \Delta^t)$$

$$\overset{(1)}{=} -\eta_l \sum_{k=0}^{K-1} \left(1 - \frac{\eta_l}{\alpha}\right)^{K-1-k} (\tilde{g}_{i,k}^t - h_i^t + \beta \Delta^t)$$

$$= -\alpha \sum_{k=0}^{K-1} \frac{\eta_l}{\alpha} \left(1 - \frac{\eta_l}{\alpha}\right)^{K-1-k} (\tilde{g}_{i,k}^t - h_i^t + \beta \Delta^t)$$

$$= -\alpha \sum_{k=0}^{K-1} \frac{\eta_l}{\alpha} \left(1 - \frac{\eta_l}{\alpha}\right)^{K-1-k} \tilde{g}_{i,k}^t + \left(1 - (1 - \frac{\eta_l}{\alpha})^K \alpha (h_i^t - \beta \Delta^t)\right)$$

$$= -\alpha \gamma \sum_{k=0}^{K-1} \frac{\gamma_k}{\gamma} \tilde{g}_{i,k}^t + \gamma \alpha (h_i^t - \beta \Delta^t).$$

(1) applies $\Psi_{i,k}^t = 0$. $\qquad\qquad \square$

**Lemma C.2.** *Under the workflow in Algorithm 1, we have:*

$$h_i^{t+1} = (1 - \gamma)h_i^t + \gamma \sum_{k=0}^{K-1} \frac{\gamma_k}{\gamma}(\tilde{g}_{i,k}^t + \beta\Delta^t) \tag{15}$$

*Proof.* According to the update rule of Line.21 in Algorithm 1, have

$$h_i^{t+1} = h_i^t - \frac{1}{\alpha}(\theta_{i,K}^t - \theta_{i,0}^t)$$

$$\overset{(1)}{=} h_i^t - \frac{1}{\alpha}(-\alpha\gamma \sum_{k=0}^{K} \frac{\gamma_k}{\gamma}\tilde{g}_{i,k}^t + \gamma\alpha(h_i^t - \beta\Delta^t))$$

$$= h_i^t + \sum_{k=0}^{K-1} \gamma_k \tilde{g}_{i,k}^t - \gamma(h_i^t - \beta\Delta^t)$$

$$= h_i^t + \frac{\eta_l}{\alpha} \sum_{k=0}^{K-1} \left(1 - \frac{\eta_l}{\alpha}\right)^{K-1-k} (\tilde{g}_{i,k}^t - h_i^t + \beta\Delta^t)$$

$$= (1 - \gamma)h_i^t + \gamma \sum_{k=0}^{K-1} \frac{\gamma_k}{\gamma}(\tilde{g}_{i,k}^t + \beta\Delta^t).$$

(1) holds due to Lemma C.1. $\qquad\square$

**Lemma C.3.** *We define the $u^{t+1} = \frac{1}{N}\sum_{i\in N}\theta_{i,K}^t$ is the averaged model among the last iteration of clients at t, the auxiliary sequence $\{z^t = u^t + \frac{1-\gamma}{\gamma}(u^t - u^{t-1})\}_{t>0}$ satisfies the rule as:*

$$z^{t+1} = z^t - \alpha\frac{1}{N}\sum_{i\in N}\sum_{k=0}^{K} \frac{\gamma_k}{\gamma}\tilde{g}_{i,k}^t - \alpha\beta\Delta^t \tag{16}$$

*Proof.* Firstly, recalling the lemma C.1 and $\theta_{i,0}^t = \theta^t = \frac{1}{N}\sum_{i\in N}(\theta_{i,K}^{t-1} - \alpha h_i^t)$ in Algorithm 1, we have:

$$u^{t+1} - u^t = \frac{1}{N}\sum_{i\in N}(\theta_{i,K}^t - \theta_{i,K}^{t-1})$$

$$= \frac{1}{N}\sum_{i\in N}(\theta_{i,K}^t - \theta_{i,0}^t - \alpha h_i^t)$$

$$= \frac{1}{N}\sum_{i\in N}(-\alpha\gamma \sum_{k=0}^{K-1} \frac{\gamma_k}{\gamma}\tilde{g}_{i,k}^t + \gamma\alpha(h_i^t - \beta\Delta^t) - \alpha h_i^t)$$

$$= -\alpha\frac{1}{N}\sum_{i\in N}\sum_{k=0}^{K-1} \frac{\gamma_k}{\gamma}(\gamma(\tilde{g}_{i,k}^t + \beta\Delta^t) + (1 - \gamma)h_i^t).$$

Here, we define a virtual observation sequence $\{u^t\}$, and its update rule is:

$$u_{i,k+1}^t = u_{i,k}^t - \alpha\frac{\gamma_k}{\gamma}(\gamma(\tilde{g}_{i,k}^t + \beta\Delta^t) + (1 - \gamma)h_i^t); \quad u_{i,0}^{t+1} = u^{t+1} = \frac{1}{N}\sum_{i\in N} u_{i,K}^t.$$

Recalling the lemma C.2 and update rule $u_{i,K}^t - u_{i,0}^t = -\alpha(1 - \gamma)h_i^t - \alpha\gamma\sum_{k=0}^{K-1}\frac{\gamma_k}{\gamma}(\tilde{g}_{i,k}^t + \beta\Delta^t)$, can get:

$$h_i^{t+1} = (1 - \gamma)h_i^t + \gamma \sum_{k=0}^{K-1} \frac{\gamma_k}{\gamma}(\tilde{g}_{i,k}^t + \beta\Delta^t)$$

$$= -\frac{1}{\alpha}(u_{i,K}^t - u_{i,0}^t) - \gamma\sum_{k=0}^{K-1}\frac{\gamma_k}{\gamma}(\tilde{g}_{i,k}^t + \beta\Delta^t) + \gamma\sum_{k=0}^{K-1}\frac{\gamma_k}{\gamma}(\tilde{g}_{i,k}^t + \beta\Delta^t)$$

$$= -\frac{1}{\alpha}(u_{i,K}^t - u_{i,0}^t).$$

Then, we can expend the auxiliary sequence $z^t$ as:

$$
\begin{aligned}
z^{t+1} - z^t &= (u^{t+1} - u^t) + \frac{1-\gamma}{\gamma}(u^{t+1} - u^t) - \frac{1-\gamma}{\gamma}(u^t - u^{t-1}) \\
&= \frac{1}{\gamma}(u^{t+1} - u^t) - \frac{1-\gamma}{\gamma}(u^t - u^{t-1}) \\
&= -\alpha\frac{1}{N}\sum_{i\in N}\left(\left(\sum_{k=0}^{K}\frac{\gamma_k}{\gamma}\left(\tilde{g}_{i,k}^t + \beta\Delta^t\right)\right) + \frac{1-\gamma}{\gamma}h_i^t\right) - \frac{1-\gamma}{\gamma}(u^t - u^{t-1}) \\
&= -\alpha\frac{1}{N}\sum_{i\in N}\sum_{k=0}^{K}\frac{\gamma_k}{\gamma}\left(\tilde{g}_{i,k}^t + \beta\Delta^t\right) - \frac{1-\gamma}{\gamma}\frac{1}{N}\sum_{i\in N}\alpha h_i^t - \frac{1-\gamma}{\gamma}(u^t - u^{t-1}) \\
&= -\alpha\frac{1}{N}\sum_{i\in N}\sum_{k=0}^{K}\frac{\gamma_k}{\gamma}\left(\tilde{g}_{i,k}^t + \beta\Delta^t\right) - \frac{1-\gamma}{\gamma}\frac{1}{N}\sum_{i\in N}(u^t - u^{t-1} + \alpha h_i^t) \\
&= -\alpha\frac{1}{N}\sum_{i\in N}\sum_{k=0}^{K}\frac{\gamma_k}{\gamma}\left(\tilde{g}_{i,k}^t + \beta\Delta^t\right) - \frac{1-\gamma}{\gamma}\frac{1}{N}\sum_{i\in N}(\theta_{i,K}^{t-1} - \theta_{i,K}^{t-2} + \alpha h_i^t) \\
&\overset{(1)}{=} -\alpha\frac{1}{N}\sum_{i\in N}\sum_{k=0}^{K}\frac{\gamma_k}{\gamma}\left(\tilde{g}_{i,k}^t + \beta\Delta^t\right) - \frac{1-\gamma}{\gamma}\frac{1}{N}\sum_{i\in N}(\theta_{i,K}^{t-1} - \theta_{i,0}^{t-1} + \alpha h_i^t - \alpha h_i^{t-1}) \\
&= -\alpha\frac{1}{N}\sum_{i\in N}\sum_{k=0}^{K}\frac{\gamma_k}{\gamma}\tilde{g}_{i,k}^t - \alpha\beta\Delta^t.
\end{aligned}
$$

(1) holds due to Line.21 in Algorithm 1. $\qquad\square$

**Lemma C.4.** *(Bounded global dual update)The global dual variable $\frac{1}{N}\sum_{i\in N}h_i^{t+1}$ holds upper bound of:*

$$
\mathbb{E}_t\left\|\frac{1}{N}\sum_{i\in N}h_i^t\right\|^2 \le \frac{1}{\gamma}\left(\mathbb{E}_t\left\|\frac{1}{N}\sum_{i\in N}h_i^t\right\|^2 - \mathbb{E}_t\left\|\frac{1}{N}\sum_{i\in N}h_i^{t+1}\right\|^2\right)
$$
$$
+ 2\mathbb{E}_t\left\|\frac{1}{N}\sum_{i\in N}\sum_{k=0}^{K-1}\frac{\gamma_k}{\gamma}\tilde{g}_{i,k}^t\right\|^2 + 2\beta^2\mathbb{E}_t\|\Delta^t\|^2. \tag{17}
$$

*Proof.* According to lemma C.2, we have:

$$
\frac{1}{N}\sum_{i\in N}h_i^{t+1} = (1-\gamma)\frac{1}{N}\sum_{i\in N}h_i^t + \gamma\frac{1}{N}\sum_{i\in N}\sum_{k=0}^{K-1}\frac{\gamma_k}{\gamma}(\tilde{g}_{i,k}^t + \beta\Delta^t).
$$

Take $L2$-norm, we have:

$$
\begin{aligned}
\left\|\frac{1}{N}\sum_{i\in N}h_i^{t+1}\right\|^2 &= \left\|(1-\gamma)\frac{1}{N}\sum_{i\in N}h_i^t + \gamma\frac{1}{N}\sum_{i\in N}\sum_{k=0}^{K-1}\frac{\gamma_k}{\gamma}(\tilde{g}_{i,k}^t + \beta\Delta^t)\right\|^2 \\
&\le (1-\gamma)\left\|\frac{1}{N}\sum_{i\in N}h_i^t\right\|^2 + \gamma\left\|\frac{1}{N}\sum_{i\in N}\sum_{k=0}^{K-1}\frac{\gamma_k}{\gamma}(\tilde{g}_{i,k}^t + \beta\Delta^t)\right\|^2 \\
&\le (1-\gamma)\left\|\frac{1}{N}\sum_{i\in N}h_i^t\right\|^2 + 2\gamma\left\|\frac{1}{N}\sum_{i\in N}\sum_{k=0}^{K-1}\frac{\gamma_k}{\gamma}\tilde{g}_{i,k}^t\right\|^2 + 2\beta^2\gamma\|\Delta^t\|^2.
\end{aligned}
$$

Take expectations. Thus, we have the following recursion:

$$\mathbb{E}_t \left\| \frac{1}{N} \sum_{i \in N} h_i^t \right\|^2 \leq \frac{1}{\gamma} \left( \mathbb{E}_t \left\| \frac{1}{N} \sum_{i \in N} h_i^t \right\|^2 - \mathbb{E}_t \left\| \frac{1}{N} \sum_{i \in N} h_i^{t+1} \right\|^2 \right)$$

$$+ 2\mathbb{E}_t \left\| \frac{1}{N} \sum_{i \in N} \sum_{k=0}^{K-1} \frac{\gamma_k}{\gamma} \tilde{g}_{i,k}^t \right\|^2 + 2\beta^2 \mathbb{E}_t \|\Delta^t\|^2. \tag{18}$$

$\square$

**Lemma C.5.** *(Bounded local dual update)The local dual variable $h_i^{t+1}$ holds upper bound of:*

$$\frac{1}{N} \sum_{i \in N} \mathbb{E}_t \|h_i^t\|^2 \leq \frac{C}{\gamma N} \sum_{i \in N} (\mathbb{E}_t \|h_i^t\|^2 - \mathbb{E}_t \|h_i^{t+1}\|^2) + 4CL^2\rho^2 + \frac{24CL^2}{N} \sum_{i \in N} \sum_{k=0}^{K-1} \frac{\gamma_k}{\gamma} \mathbb{E}_t \left\| \theta_{i,k}^t - \theta^t \right\|^2$$

$$+ (12 + 2\beta^2)C\mathbb{E}_t \left\| \nabla f(z^t) \right\|^2 + 2C(6\sigma_g^2 + \sigma_l^2). \tag{19}$$

*where $\frac{1}{C} = 1 - \frac{24\alpha^2 L^2 (1-2\gamma)^2}{\gamma^2}$ is the constant.*

*Proof.* Recalling lemma C.2,

$$h_i^{t+1} = (1 - \gamma)h_i^t + \gamma \sum_{k=0}^{K-1} \frac{\gamma_k}{\gamma} (\tilde{g}_{i,k}^t + \beta \Delta^t).$$

same like lemma C.4's proof, we have:

$$\frac{1}{N} \sum_{i \in N} \mathbb{E}_t \|h_i^t\|^2 \leq \frac{1}{\gamma N} \sum_{i \in N} (\mathbb{E}_t \|h_i^t\|^2 - \mathbb{E}_t \|h_i^{t+1}\|^2) + \frac{2}{N} \sum_{i \in N} \sum_{k=0}^{K-1} \frac{\gamma_k}{\gamma} \mathbb{E}_t \|\tilde{g}_{i,k}^t\|^2 + 2\beta^2 \mathbb{E}_t \|\Delta^t\|^2.$$

Here, we provide an upper bound for the quasi-stochastic gradient:

$$\frac{1}{N} \sum_{i \in N} \sum_{k=0}^{K-1} \frac{\gamma_k}{\gamma} \mathbb{E}_t \|\tilde{g}_{i,k}^t\|^2$$

$$= \frac{1}{N} \sum_{i \in N} \sum_{k=0}^{K-1} \frac{\gamma_k}{\gamma} \mathbb{E}_t \left\| \nabla f_i(\theta_{i,k}^t + \delta_{i,k}^t) \right\|^2 + \sigma_l^2$$

$$= \frac{1}{N} \sum_{i \in N} \sum_{k=0}^{K-1} \frac{\gamma_k}{\gamma} \mathbb{E}_t \left\| \nabla f_i(\theta_{i,k}^t + \delta_{i,k}^t) - \nabla f_i(\theta_{i,k}^t) + \nabla f_i(\theta_{i,k}^t) \right\|^2 + \sigma_l^2$$

$$\leq 2L^2\rho^2 + \frac{2}{N} \sum_{i \in N} \sum_{k=0}^{K-1} \frac{\gamma_k}{\gamma} \mathbb{E}_t \left\| \nabla f_i(\theta_{i,k}^t) - \nabla f_i(z^t) + \nabla f_i(z^t) - \nabla f(z^t) + \nabla f(z^t) \right\|^2 + \sigma_l^2$$

$$\leq 2L^2\rho^2 + \frac{6}{N} \sum_{i \in N} \sum_{k=0}^{K-1} \frac{\gamma_k}{\gamma} \mathbb{E}_t \left\| \theta_{i,k}^t - z^t \right\|^2 + 6\mathbb{E}_t \left\| \nabla f(z^t) \right\|^2 + (6\sigma_g^2 + \sigma_l^2)$$

$$\leq 2L^2\rho^2 + \frac{6L^2}{N} \sum_{i \in N} \sum_{k=0}^{K-1} \frac{\gamma_k}{\gamma} \mathbb{E}_t \left\| \theta_{i,k}^t - \theta^t + \theta^t - u^t + u^t - z^t \right\|^2 + 6\mathbb{E}_t \left\| \nabla f(z^t) \right\|^2 + (6\sigma_g^2 + \sigma_l^2)$$

$$\leq 2L^2\rho^2 + \frac{12L^2}{N} \sum_{i \in N} \sum_{k=0}^{K-1} \frac{\gamma_k}{\gamma} \mathbb{E}_t \left\| \theta_{i,k}^t - \theta^t \right\|^2 + 12L^2 \left\| \theta^t - u^t + u^t - z^t \right\|^2 + 6\mathbb{E}_t \left\| \nabla f(z^t) \right\|^2 + (6\sigma_g^2 + \sigma_l^2)$$

$$\overset{(1)}{\leq} 2L^2\rho^2 + \frac{12L^2}{N} \sum_{i \in N} \sum_{k=0}^{K-1} \frac{\gamma_k}{\gamma} \mathbb{E}_t \left\| \theta_{i,k}^t - \theta^t \right\|^2 + 12L^2 \frac{1}{N} \sum_{i \in N} \left\| -\alpha h_i^t + \frac{1-\gamma}{\gamma} \alpha h_i^t \right\|^2$$

$$+ 6\mathbb{E}_t \left\| \nabla f(z^t) \right\|^2 + (6\sigma_g^2 + \sigma_l^2)$$

$$\leq 2L^2\rho^2 + \frac{12L^2}{N}\sum_{i\in N}\sum_{k=0}^{K-1}\frac{\gamma_k}{\gamma}\mathbb{E}_t\left\|\theta_{i,k}^t - \theta^t\right\|^2 + \frac{12\alpha^2 L^2(1-2\gamma)^2}{\gamma^2 N}\sum_{i\in N}\mathbb{E}_t\left\|h_i^t\right\|^2$$

$$+ 6\mathbb{E}_t\left\|\nabla f(z^t)\right\|^2 + (6\sigma_g^2 + \sigma_l^2). \tag{20}$$

Inequality (1) holds because $u^t - z^t = -\frac{1-\gamma}{\gamma}(u^t - u^{t-1}); \theta^t - u^t = -\alpha\frac{1}{N}\sum_{i\in N}h_i^t$. Let $\frac{1}{C} = 1 - \frac{24\alpha^2 L^2(1-2\gamma)^2}{\gamma^2}$ is the constant. Combining the above inequalities, we have:

$$\frac{1}{N}\sum_{i\in N}\mathbb{E}_t\|h_i^t\|^2 \leq \frac{C}{\gamma N}\sum_{i\in N}(\mathbb{E}_t\|h_i^t\|^2 - \mathbb{E}_t\|h_i^{t+1}\|^2) + 4CL^2\rho^2 + \frac{24CL^2}{N}\sum_{i\in N}\sum_{k=0}^{K-1}\frac{\gamma_k}{\gamma}\mathbb{E}_t\left\|\theta_{i,k}^t - \theta^t\right\|^2$$

$$+ (12 + 2\beta^2)C\mathbb{E}_t\left\|\nabla f(z^t)\right\|^2 + 2C(6\sigma_g^2 + \sigma_l^2).$$

We set $\Delta^t \approx \nabla f(z^t)$ like Xu et al. (2021); Qu et al. (2022); Fan et al. (2024a). $\qquad\square$

Now we have completed all the preparations for the proof of Theorem 2. For the non-convex case, based on assumptions 1-3, we take the conditional expectation at round $t+1$ and expand the $f(z^{t+1})$ as:

$$\mathbb{E}_t f(z^{t+1})$$

$$\leq \mathbb{E}_t f(z^t) + \mathbb{E}_t\langle\nabla f(z^t), z^{t+1} - z^t\rangle + \frac{L}{2}\mathbb{E}_t\|z^{t+1} - z^t\|^2$$

$$= \mathbb{E}_t f(z^t) + \mathbb{E}_t\left\langle\nabla f(z^t), -\alpha\frac{1}{N}\sum_{i\in N}\sum_{k=0}^{K}\frac{\gamma_k}{\gamma}\left(\tilde{g}_{i,k}^t + \beta\nabla f(z^t)\right)\right\rangle + \frac{L}{2}\mathbb{E}_t\|z^{t+1} - z^t\|^2$$

$$= \mathbb{E}_t f(z^t) - \alpha\mathbb{E}_t\left\langle\nabla f(z^t), \frac{1}{N}\sum_{i\in N}\sum_{k=0}^{K}\frac{\gamma_k}{\gamma}\tilde{g}_{i,k}^t + \beta\nabla f(z^t) - \nabla f(z^t) + \nabla f(z^t)\right\rangle + \frac{L}{2}\mathbb{E}_t\|z^{t+1} - z^t\|^2$$

$$= \mathbb{E}_t f(z^t) - \alpha(1+\beta)\|\nabla f(z^t)\|^2 \underbrace{-\alpha\mathbb{E}_t\left\langle\nabla f(z^t), \frac{1}{N}\sum_{i\in N}\sum_{k=0}^{K}\frac{\gamma_k}{\gamma}\tilde{g}_{i,k}^t - \nabla f(z^t)\right\rangle}_{\textbf{A.1}} + \frac{L}{2}\underbrace{\mathbb{E}_t\|z^{t+1} - z^t\|^2}_{\textbf{A.2}}.$$

$$\tag{21}$$

Firstly, the term **A.1** can be bounded:

$$\textbf{A.1} = -\alpha\mathbb{E}_t\left\langle\nabla f(z^t), \frac{1}{N}\sum_{i\in N}\sum_{k=0}^{K}\frac{\gamma_k}{\gamma}\tilde{g}_{i,k}^t - \nabla f(z^t)\right\rangle$$

$$\stackrel{(1)}{=} -\alpha\mathbb{E}_t\left\langle\nabla f(z^t), \frac{1}{N}\sum_{i\in N}\sum_{k=0}^{K}\frac{\gamma_k}{\gamma}\tilde{g}_{i,k}^t - \frac{1}{N}\sum_{i\in N}\sum_{k=0}^{K}\frac{\gamma_k}{\gamma}\nabla f_i(z^t)\right\rangle$$

$$\stackrel{(2)}{=} \frac{\alpha}{2}\|\nabla f(z^t)\|^2 + \frac{\alpha}{2}\mathbb{E}_t\left\|\frac{1}{N}\sum_{i\in N}\sum_{k=0}^{K}\frac{\gamma_k}{\gamma}\left(\mathbb{E}\tilde{g}_{i,k}^t - \nabla f_i(z^t)\right)\right\|^2 - \frac{\alpha}{2N^2}\mathbb{E}_t\left\|\sum_{i\in N}\sum_{k=0}^{K-1}\frac{\gamma_k}{\gamma}\mathbb{E}\tilde{g}_{i,k}^t\right\|^2$$

$$\stackrel{(3)}{\leq} \frac{\alpha}{2}\|\nabla f(z^t)\|^2 + \frac{\alpha}{2}\frac{1}{N}\sum_{i\in N}\sum_{k=0}^{K}\frac{\gamma_k}{\gamma}\mathbb{E}_t\left\|\mathbb{E}\tilde{g}_{i,k}^t - \nabla f_i(z^t)\right\|^2 - \frac{\alpha}{2N^2}\mathbb{E}_t\left\|\sum_{i\in N}\sum_{k=0}^{K-1}\frac{\gamma_k}{\gamma}\mathbb{E}\tilde{g}_{i,k}^t\right\|^2.$$

$$\tag{22}$$

(1) holds due to the fact $\frac{1}{N}\sum_{i\in N}\nabla f_i(z^t) = \nabla f(z^t)$. (b) applies $-\langle x, y\rangle = \frac{1}{2}(\|x\|^2 + \|y\|^2 - \|x+y\|^2)$ (c) holds due to Jensen's inequality. And, according SAM update rule we have $\mathbb{E}\tilde{g}_{i,k}^t = \nabla f(\theta_{i,k}^t + \delta_{i,k}^t)$. Then, we can bounded the term $\frac{1}{N}\sum_{i\in N}\sum_{k=0}^{K}\frac{\gamma_k}{\gamma}\mathbb{E}_t\left\|\mathbb{E}\tilde{g}_{i,k}^t - \nabla f_i(z^t)\right\|^2$ as

follows:

$$\frac{1}{N}\sum_{i \in N}\sum_{k=0}^{K}\frac{\gamma_k}{\gamma}\mathbb{E}_t\left\|\mathbb{E}\tilde{g}_{i,k}^t - \nabla f_i(z^t)\right\|^2$$

$$\leq \frac{1}{N}\sum_{i \in N}\sum_{k=0}^{K}\frac{\gamma_k}{\gamma}\mathbb{E}_t\left\|\nabla f(\theta_{i,k}^t + \delta_{i,k}^t) - \nabla f_i(z^t)\right\|^2$$

$$= \frac{1}{N}\sum_{i \in N}\sum_{k=0}^{K}\frac{\gamma_k}{\gamma}\mathbb{E}_t\left\|\nabla f(\theta_{i,k}^t + \delta_{i,k}^t) - f(\theta_{i,k}^t) + f(\theta_{i,k}^t) - \nabla f_i(z^t)\right\|^2$$

$$\leq 2L^2\rho^2 + \frac{2L^2}{N}\sum_{i \in N}\sum_{k=0}^{K}\frac{\gamma_k}{\gamma}\mathbb{E}_t\left\|\theta_{i,k}^t - z^t\right\|^2$$

$$= \frac{2L^2}{N}\sum_{i \in N}\sum_{k=0}^{K}\frac{\gamma_k}{\gamma}\mathbb{E}_t\left\|\theta_{i,k}^t - \theta^t + \theta^t - u^t + u^t - z^t\right\|^2 + 2L^2\rho^2$$

$$\leq \frac{4L^2}{N}\sum_{i \in N}\sum_{k=0}^{K}\frac{\gamma_k}{\gamma}\mathbb{E}_t\left\|\theta_{i,k}^t - \theta^t\right\|^2 + 4L^2\frac{\gamma_k}{\gamma}\mathbb{E}_t\left\|\theta^t - u^t + u^t - z^t\right\|^2 + 2L^2\rho^2$$

$$= \frac{4L^2}{N}\sum_{i \in N}\sum_{k=0}^{K}\frac{\gamma_k}{\gamma}\mathbb{E}_t\left\|\theta_{i,k}^t - \theta^t\right\|^2 + 4L^2\frac{\gamma_k}{\gamma}\mathbb{E}_t\left\|-\alpha\frac{1}{N}\sum_{i \in N}h_i^t + \frac{\gamma-1}{\gamma}(u^{t-1} - u^{t-1})\right\|^2 + 2L^2\rho^2$$

$$= \frac{4L^2}{N}\sum_{i \in N}\sum_{k=0}^{K}\frac{\gamma_k}{\gamma}\mathbb{E}_t\left\|\theta_{i,k}^t - \theta^t\right\|^2 + \frac{4\alpha^2 L^2(1-2\gamma)^2}{\gamma^2}\mathbb{E}_t\left\|\frac{1}{N}\sum_{i \in N}h_i^t\right\|^2 + 2L^2\rho^2$$

$$\overset{(1)}{\leq} \frac{4L^2}{N}\sum_{i \in N}\sum_{k=0}^{K}\frac{\gamma_k}{\gamma}\mathbb{E}_t\left\|\theta_{i,k}^t - \theta^t\right\|^2 + \frac{4\alpha^2 L^2(1-2\gamma)^2}{\gamma^3}\left(\mathbb{E}_t\left\|\frac{1}{N}\sum_{i \in N}h_i^t\right\|^2 - \mathbb{E}_t\left\|\frac{1}{N}\sum_{i \in N}h_i^{t+1}\right\|^2\right)$$

$$+ \frac{8\alpha^2 L^2(1-2\gamma)^2}{\gamma^2}\mathbb{E}_t\left\|\frac{1}{N}\sum_{i \in N}\sum_{k=0}^{K-1}\frac{\gamma_k}{\gamma}\tilde{g}_{i,k}^t\right\|^2 + \frac{8\alpha^2 L^2(1-2\gamma)^2}{\gamma^2}\beta^2\mathbb{E}_t\|\nabla f(z^t)\|^2 + 2L^2\rho^2.$$

$$(23)$$

(1) applied the lemma C.4.

Then, we assume $\epsilon^t = \frac{1}{N}\sum_{i \in N}\sum_{k=0}^{K-1}\frac{\gamma_k}{\gamma}\mathbb{E}_t\|\theta_{i,k}^t - \theta^t\|^2$ term as the local offset after $k$ iterations. we first bounded $\epsilon_k^t = \frac{1}{N}\sum_{i \in N}\mathbb{E}_t\|\theta_{i,k}^t - \theta^t\|^2$ as:

$$\epsilon_k^t = \frac{1}{N}\sum_{i \in N}\mathbb{E}_t\|\theta_{i,k}^t - \theta^t\|^2 = \frac{1}{N}\sum_{i \in N}\mathbb{E}_t\|\theta_{i,k}^t - \theta_{i,k-1}^t + \theta_{i,k-1}^t - \theta_{i,0}^t\|^2$$

$$\overset{(1)}{=} \frac{1}{N}\sum_{i \in N}\left\|-\eta_l(\tilde{g}_{i,k-1}^t - h_i^t) + \left(1 - \frac{\eta_l}{\alpha}\right)(\theta_{i,k-1}^t - \theta_{i,0}^t) - \eta_l\beta\Delta^t\right\|^2$$

$$\overset{(2)}{\leq} (1+b)\left(1 - \frac{\eta_l}{\alpha}\right)^2\frac{1}{N}\sum_{i \in N}\mathbb{E}_t\|\theta_{i,k-1}^t - \theta_{i,0}^t\|^2 + (1+\frac{1}{b})\frac{\eta_l^2}{N}\sum_{i \in N}\mathbb{E}_t\|\tilde{g}_{i,k-1}^t - h_i^t + \beta\Delta^t\|^2$$

$$= (1+b)\left(1 - \frac{\eta_l}{\alpha}\right)^2\epsilon_{k-1}^t + (1+\frac{1}{\alpha})\frac{\eta_l^2}{N}\sum_{i \in N}\mathbb{E}_t\|\nabla f_i(\theta_{i,k-1}^t + \delta_{i,k-1}^t) - h_i^t + \beta\Delta^t\|^2 + (1+\frac{1}{b})\eta_l^2\sigma_l^2$$

$$\leq (1+\frac{1}{b})\frac{3\eta_l^2}{N}\sum_{i \in N}\left(\mathbb{E}_t\|\nabla f_i(\theta_{i,k-1}^t + \delta_{i,k-1}^t)\|^2 + \mathbb{E}_t\|h_i^t\|^2 + \mathbb{E}_t\|\beta\Delta^t\|^2\right)$$

$$+ (1+\frac{1}{b})\eta_l^2\sigma_l^2 + (1+b)\left(1 - \frac{\eta_l}{\alpha}\right)^2\epsilon_{k-1}^t$$

$$= (1+\frac{1}{b})\frac{3\eta_l^2}{N}\sum_{i \in N}\mathbb{E}_t\|\nabla f_i(\theta_{i,k-1}^t + \delta_{i,k-1}^t) - \nabla f_i(\theta_{i,k-1}^t) + \nabla f_i(\theta_{i,k-1}^t)\|^2$$

$$+ (1 + \frac{1}{b})\frac{3\eta_l^2}{N}\sum_{i \in N}\mathbb{E}_t\|h_i^t\|^2 + (1 + \frac{1}{b})3\eta_l^2\beta^2\mathbb{E}_t\|\nabla f(z^t)\|^2$$

$$+ (1 + \frac{1}{b})\eta_l^2\sigma_l^2 + (1 + b)\left(1 - \frac{\eta_l}{\alpha}\right)^2\epsilon_{k-1}^t$$

$$\leq (1 + \frac{1}{b})\frac{6\eta_l^2}{N}\sum_{i \in N}\mathbb{E}_t\|\nabla f_i(\theta_{i,k-1}^t) - \nabla f_i(\theta^t) + \nabla f_i(\theta^t) - \nabla f_i(z^t) + \nabla f_i(z^t) - \nabla f(z^t) + \nabla f(z^t)\|^2$$

$$+ (1 + \frac{1}{b})\frac{3\eta_l^2}{N}\sum_{i \in N}\mathbb{E}_t\|h_i^t\|^2 + (1 + \frac{1}{b})3\eta_l^2\beta^2\mathbb{E}_t\|\nabla f(z^t)\|^2 + (1 + \frac{1}{b})\eta_l^2(\sigma_l^2 + 6L^2\rho^2)$$

$$+ (1 + b)\left(1 - \frac{\eta_l}{\alpha}\right)^2\epsilon_{k-1}^t$$

$$\leq (1 + \frac{1}{b})\frac{24\eta_l^2 L^2}{N}\sum_{i \in N}\mathbb{E}_t\|\theta_{i,k-1}^t - \theta^t\|^2 + (1 + \frac{1}{b})24\eta_l^2 L^2\|\theta^t - u^t + u^t - z^t\|^2 + (1 + \frac{1}{b})24\eta_l^2\|\nabla f(z^t)\|^2$$

$$+ (1 + \frac{1}{b})\frac{3\eta_l^2}{N}\sum_{i \in N}\mathbb{E}_t\|h_i^t\|^2 + (1 + \frac{1}{b})3\eta_l^2\beta^2\mathbb{E}_t\|\nabla f(z^t)\|^2$$

$$+ (1 + \frac{1}{b})\eta_l^2(\sigma_l^2 + 6L^2\rho^2 + 24\sigma_g^2) + (1 + b)\left(1 - \frac{\eta_l}{\alpha}\right)^2\epsilon_{k-1}^t$$

$$\overset{(3)}{\leq} \left((1 + b)\left(1 - \frac{\eta_l}{\alpha}\right)^2 + (1 + \frac{1}{b})24\eta_l^2 L^2\right)\epsilon_{k-1}^t + (1 + \frac{1}{b})\eta_l^2\left(\frac{24L^2\alpha^2(1 - 2\gamma)^2}{\gamma^2} + 3\right)\frac{1}{N}\sum_{i \in N}\mathbb{E}_t\|h_i^t\|^2$$

$$+ (1 + \frac{1}{b})3\eta_l^2(8 + \beta^2)\mathbb{E}_t\|\nabla f(z^t)\|^2 + (1 + \frac{1}{b})\eta_l^2(\sigma_l^2 + 6L^2\rho^2 + 24\sigma_g^2)$$

$$\leq \left((1 + b)\left(1 - \frac{\eta_l}{\alpha}\right)^2 + (1 + \frac{1}{b})24\eta_l^2 L^2\right)\epsilon_{k-1}^t + (1 + \frac{1}{b})\eta_l^2(\sigma_l^2 + 6L^2\rho^2 + 24\sigma_g^2)$$

$$+ (1 + \frac{1}{b})\eta_l^2\left(\frac{24L^2\alpha^2(1 - 2\gamma)^2}{\gamma^2} + 3\right)\left(\frac{C}{\gamma N}\sum_{i \in N}(\mathbb{E}_t\|h_i^t\|^2 - \mathbb{E}_t\|h_i^{t+1}\|^2) + 4CL^2\rho^2 + 2C(6\sigma_g^2 + \sigma_l^2)\right.$$

$$+ \frac{24CL^2}{N}\sum_{i \in N}\sum_{k=0}^{K-1}\frac{\gamma_k}{\gamma}\mathbb{E}_t\left\|\theta_{i,k}^t - \theta^t\right\|^2 + (12 + 2\beta^2)C\mathbb{E}_t\left\|\nabla f(z^t)\right\|^2\Big) + (1 + \frac{1}{b})3\eta_l^2(8 + \beta^2)\mathbb{E}_t\|\nabla f(z^t)\|^2$$

$$= \left((1 + b)\left(1 - \frac{\eta_l}{\alpha}\right)^2 + (1 + \frac{1}{b})24\eta_l^2 L^2\right)\epsilon_{k-1}^t + (1 + \frac{1}{b})\eta_l^2(\sigma_l^2 + 6L^2\rho^2 + 24\sigma_g^2)$$

$$+ (1 + \frac{1}{b})\eta_l^2\left(\frac{4C - 1}{C}\right)\left(\frac{C}{\gamma N}\sum_{i \in N}(\mathbb{E}_t\|h_i^t\|^2 - \mathbb{E}_t\|h_i^{t+1}\|^2) + 4CL^2\rho^2 + 2C(6\sigma_g^2 + \sigma_l^2)\right.$$

$$+ \frac{24CL^2}{N}\sum_{i \in N}\sum_{k=0}^{K-1}\frac{\gamma_k}{\gamma}\mathbb{E}_t\left\|\theta_{i,k}^t - \theta^t\right\|^2 + (12 + 2\beta^2)C\mathbb{E}_t\left\|\nabla f(z^t)\right\|^2\Big) + (1 + \frac{1}{b})3\eta_l^2(8 + \beta^2)\mathbb{E}_t\|\nabla f(z^t)\|^2$$

$$= \left((1 + b)\left(1 - \frac{\eta_l}{\alpha}\right)^2 + (1 + \frac{1}{b})24\eta_l^2 L^2\right)\epsilon_{k-1}^t + (1 + \frac{1}{b})\eta_l^2(\sigma_l^2 + 6L^2\rho^2 + 24\sigma_g^2)$$

$$+ (1 + \frac{1}{b})\eta_l^2\frac{4C - 1}{\gamma N}\sum_{i \in N}(\mathbb{E}_t\|h_i^t\|^2 - \mathbb{E}_t\|h_i^{t+1}\|^2) + (1 + \frac{1}{b})\eta_l^2(16C - 4)L^2\rho^2 + (1 + \frac{1}{b})\eta_l^2(8C - 2)(6\sigma_g^2 + \sigma_l^2)$$

$$+ (1 + \frac{1}{b})\eta_l^2(96C - 24)L^2\epsilon^t + (1 + \frac{1}{b})\eta_l^2(12 + 2\beta^2)(4C - 1)\mathbb{E}_t\left\|\nabla f(z^t)\right\|^2 + (1 + \frac{1}{b})3\eta_l^2(8 + \beta^2)\mathbb{E}_t\|\nabla f(z^t)\|^2$$

$$\overset{(4)}{\leq} \left((1 + b)\left(1 - \frac{\eta_l}{\alpha}\right)^2 + (1 + \frac{1}{b})24\eta_l^2 L^2\right)\epsilon_{k-1}^t + (1 + \frac{1}{b})\eta_l^2(\sigma_l^2 + 6L^2\rho^2 + 24\sigma_g^2)$$

$$+ (1 + \frac{1}{b})\frac{7\eta_l^2}{\gamma N}\sum_{i \in N}(\mathbb{E}_t\|h_i^t\|^2 - \mathbb{E}_t\|h_i^{t+1}\|^2) + 14(1 + \frac{1}{b})\eta_l^2(\sigma_l^2 + 2L^2\rho^2 + 6\sigma_g^2)$$

$$+ 168(1 + \frac{1}{b})\eta_l^2 L^2\epsilon^t + (1 + \frac{1}{b})7\eta_l^2(12 + 2\beta^2)\mathbb{E}_t\left\|\nabla f(z^t)\right\|^2 + (1 + \frac{1}{b})3\eta_l^2(8 + \beta^2)\mathbb{E}_t\|\nabla f(z^t)\|^2.$$

(1) holds due to Line.19 in Algorithm 1, (2) uses the fact $\|x + y\|^2 \leq (1 + b)\|x\|^2 + (1 + \frac{1}{b})\|y\|^2$, (3) applies lemma C.5, (4) applies $C$ satisfies $C \leq 2$, which means $\left(\frac{24L^2\alpha^2(1-2\gamma)^2}{\gamma^2} + 3\right) = \frac{4C-1}{C}$, $\frac{1}{C} = 1 - \frac{24\alpha^2 L^2(1-2\gamma)^2}{\gamma^2} \geq \frac{1}{2}$.

We let the weight satisfy thatSun et al. (2023c):

$$(1 + b)\left(1 - \frac{\eta_l}{\alpha}\right)^2 + (1 + \frac{1}{b})24\eta_l^2 L^2 \leq \frac{\gamma_{K-2}}{\gamma_{K-1}} = \frac{\gamma_{K-3}}{\gamma_{K-2}} = \cdots = \frac{\gamma_1}{\gamma_0} = 1 - \frac{\eta_l}{\alpha} \tag{24}$$

let $\eta_l \leq \alpha$, we have:

$$\epsilon^t = \sum_{k=0}^{K-1} \frac{\gamma_k}{\gamma} \epsilon_k^t$$

$$\leq 7(1 + \frac{1}{b})\frac{\eta_l^2}{\gamma} \sum_{\check{k}=0}^{K-1} \left(\sum_{k=0}^{\check{k}-1} \gamma_k\right) \left(3\sigma_l^2 + 5L^2\rho^2 + 16\sigma_g^2 + 24\epsilon^t + (16 + 3\beta^2)\mathbb{E}_t\|\nabla f(z^t)\|^2 \right.$$

$$\left. + \frac{1}{\gamma N} \sum_{i \in N} (\mathbb{E}_t\|h_i^t\|^2 - \mathbb{E}_t\|h_i^{t+1}\|^2)\right)$$

$$\leq 7(1 + \frac{1}{b})\eta_l^2 \sum_{\check{k}=0}^{K-1} \left(\sum_{k=0}^{K-1} \frac{\gamma_k}{\gamma}\right) \left(3\sigma_l^2 + 5L^2\rho^2 + 16\sigma_g^2 + 16\epsilon^t + (16 + 3\beta^2)\mathbb{E}_t\|\nabla f(z^t)\|^2 \right.$$

$$\left. + \frac{1}{\gamma N} \sum_{i \in N} (\mathbb{E}_t\|h_i^t\|^2 - \mathbb{E}_t\|h_i^{t+1}\|^2)\right)$$

$$= 7(1 + \frac{1}{b})\eta_l^2 K \left(3\sigma_l^2 + 5L^2\rho^2 + 16\sigma_g^2 + (16 + 3\beta^2)\mathbb{E}_t\|\nabla f(z^t)\|^2 + \frac{1}{\gamma N} \sum_{i \in N} (\mathbb{E}_t\|h_i^t\|^2 - \mathbb{E}_t\|h_i^{t+1}\|^2)\right)$$

$$+ 168(1 + \frac{1}{b})\eta_l^2 K L^2 \epsilon^t. \tag{25}$$

Let $\eta_l$ satisfies the bound of $\eta_l \leq \frac{1}{\sqrt{336(1+1/b)KL}}$ for convenience, we can bound the $\epsilon^t$ as:

$$\epsilon^t \leq 14(1 + \frac{1}{b})\eta_l^2 K \left(3\sigma_l^2 + 5L^2\rho^2 + 16\sigma_g^2 + (16 + 3\beta^2)\mathbb{E}_t\|\nabla f(z^t)\|^2 + \frac{1}{\gamma N} \sum_{i \in N} (\mathbb{E}_t\|h_i^t\|^2 - \mathbb{E}_t\|h_i^{t+1}\|^2)\right). \tag{26}$$

Let $b = 1$ for convenience, we can get:

$$\frac{1}{N} \sum_{i \in N} \sum_{k=0}^{K} \frac{\gamma_k}{\gamma} \mathbb{E}_t \left\|\mathbb{E}\tilde{g}_{i,k}^t - \nabla f_i(z^t)\right\|^2$$

$$\leq 4L^2\epsilon^t + \frac{4\alpha^2 L^2(1-2\gamma)^2}{\gamma^3} \left(\mathbb{E}_t \left\|\frac{1}{N} \sum_{i \in N} h_i^t\right\|^2 - \mathbb{E}_t \left\|\frac{1}{N} \sum_{i \in N} h_i^{t+1}\right\|^2\right)$$

$$+ \frac{8\alpha^2 L^2(1-2\gamma)^2}{\gamma^2} \mathbb{E}_t \left\|\frac{1}{N} \sum_{i \in N} \sum_{k=0}^{K-1} \frac{\gamma_k}{\gamma} \tilde{g}_{i,k}^t\right\|^2 + \frac{8\alpha^2 L^2(1-2\gamma)^2}{\gamma^2} \beta^2 \mathbb{E}_t\|\nabla f(z^t)\|^2 + 2L^2\rho^2$$

$$\leq \frac{4\alpha^2 L^2(1-2\gamma)^2}{\gamma^3} \left(\mathbb{E}_t \left\|\frac{1}{N} \sum_{i \in N} h_i^t\right\|^2 - \mathbb{E}_t \left\|\frac{1}{N} \sum_{i \in N} h_i^{t+1}\right\|^2\right) + 2L^2\rho^2$$

$$+ \frac{8\alpha^2 L^2 (1-2\gamma)^2}{\gamma^2} \mathbb{E}_t \left\| \frac{1}{N} \sum_{i \in N} \sum_{k=0}^{K-1} \frac{\gamma_k}{\gamma} \tilde{g}_{i,k}^t \right\|^2 + \frac{8\alpha^2 L^2 (1-2\gamma)^2}{\gamma^2} \beta^2 \mathbb{E}_t \|\nabla f(z^t)\|^2$$

$$+ \frac{112 L^2 \eta_l^2 K}{\gamma N} \sum_{i \in N} \left( \mathbb{E}_t \|h_i^t\|^2 - \mathbb{E}_t \|h_i^{t+1}\|^2 \right) + 112\eta_l^2 L^2 K (3\sigma_l^2 + 5L^2\rho^2 + 16\sigma_g^2)$$

$$+ 112\eta_l^2 L^2 K (16 + 3\beta^2) \|\nabla f(z^t)\|^2. \tag{27}$$

Thus we can bound the **A.1** as follow:

$$\mathbf{A.1} \leq \frac{\alpha}{2} \mathbb{E}_t \|\nabla f(z^t)\|^2 + \frac{\alpha}{2} \frac{1}{N} \sum_{i \in N} \sum_{k=0}^{K} \frac{\gamma_k}{\gamma} \mathbb{E}_t \left\| \mathbb{E}\tilde{g}_{i,k}^t - \nabla f_i(z^t) \right\|^2 - \frac{\alpha}{2N^2} \mathbb{E}_t \left\| \sum_{i \in N} \sum_{k=0}^{K-1} \frac{\gamma_k}{\gamma} \mathbb{E}\tilde{g}_{i,k}^t \right\|^2$$

$$\leq \left( \frac{\alpha}{2} + 896\alpha\eta_l^2 L^2 K + 168\alpha\eta_l^2 L^2 K\beta^2 + \alpha\beta^2 \right) \mathbb{E}_t \|\nabla f(z^t)\|^2 + \frac{56\alpha L^2 \eta_l^2 K}{\gamma N} \sum_{i \in N} \left( \mathbb{E}_t \|h_i^t\|^2 - \mathbb{E}_t \|h_i^{t+1}\|^2 \right)$$

$$+ \frac{2\alpha^3 L^2 (1-2\gamma)^2}{\gamma^3} \left( \mathbb{E}_t \left\| \frac{1}{N} \sum_{i \in N} h_i^t \right\|^2 - \mathbb{E}_t \left\| \frac{1}{N} \sum_{i \in N} h_i^{t+1} \right\|^2 \right) + \alpha L^2 \rho^2 - \frac{\alpha}{2N^2} \mathbb{E}_t \left\| \sum_{i \in N} \sum_{k=0}^{K-1} \frac{\gamma_k}{\gamma} \mathbb{E}\tilde{g}_{i,k}^t \right\|^2$$

$$+ \frac{4\alpha^3 L^2 (1-2\gamma)^2}{\gamma^2} \mathbb{E}_t \left\| \frac{1}{N} \sum_{i \in N} \sum_{k=0}^{K-1} \frac{\gamma_k}{\gamma} \tilde{g}_{i,k}^t \right\|^2 + 56\alpha\eta_l^2 L^2 K (3\sigma_l^2 + 16\sigma_g^2 + 5L^2\rho^2). \tag{28}$$

We notice that **A.1** contains the term $\mathbb{E}_t \left\| \frac{1}{N} \sum_{i \in N} \sum_{k=0}^{K-1} \frac{\gamma_k}{\gamma} \tilde{g}_{i,k}^t \right\|^2$ with a negative weight, thus we can set a suitable $\alpha$ to eliminate this term. Besides, the upper bound of **A.2** can be easy to get:

$$\mathbf{A.2} = \mathbb{E}_t \|z^{t+1} - z^t\|^2$$

$$= \mathbb{E}_t \left\| \alpha \frac{1}{N} \sum_{i \in N} \sum_{k=0}^{K} \frac{\gamma_k}{\gamma} \tilde{g}_{i,k}^t + \alpha\beta\Delta^t \right\|^2$$

$$\leq \frac{2\alpha^2}{N^2} \mathbb{E}_t \left\| \sum_{i \in N} \sum_{k=0}^{K} \frac{\gamma_k}{\gamma} \tilde{g}_{i,k}^t \right\|^2 + 2\alpha^2\beta^2 \mathbb{E}_t \left\| \Delta^t \right\|^2. \tag{29}$$

As we have bounded the term **A.1** and **A.2**, we combine the inequalities above and get:

$$\mathbb{E}_t f(z^{t+1})$$

$$\leq \mathbb{E}_t f(z^t) - \alpha(1+\beta)\|\nabla f(z^t)\|^2 + \mathbf{A.1} + \frac{L}{2}\mathbf{A.2}$$

$$\leq \mathbb{E}_t f(z^t) - \alpha(1+\beta)\|\nabla f(z^t)\|^2 + \frac{L\alpha^2}{N^2} \mathbb{E}_t \left\| \sum_{i \in N} \sum_{k=0}^{K} \frac{\gamma_k}{\gamma} \tilde{g}_{i,k}^t \right\|^2 + L\alpha^2\beta^2 \mathbb{E}_t \left\| \nabla f(z^t) \right\|^2$$

$$- \left( \frac{\alpha}{2} + \alpha\beta - 896\alpha\eta_l^2 L^2 K - 168\alpha\eta_l^2 L^2 K\beta^2 - \alpha\beta^2 \right) \mathbb{E}_t \|\nabla f(z^t)\|^2 + \frac{56\alpha L^2 \eta_l^2 K}{\gamma N} \sum_{i \in N} \left( \mathbb{E}_t \|h_i^t\|^2 - \mathbb{E}_t \|h_i^{t+1}\|^2 \right)$$

$$+ \frac{2\alpha^3 L^2 (1-2\gamma)^2}{\gamma^3} \left( \mathbb{E}_t \left\| \frac{1}{N} \sum_{i \in N} h_i^t \right\|^2 - \mathbb{E}_t \left\| \frac{1}{N} \sum_{i \in N} h_i^{t+1} \right\|^2 \right) + \alpha L^2 \rho^2 - \frac{\alpha}{2N^2} \mathbb{E}_t \left\| \sum_{i \in N} \sum_{k=0}^{K-1} \frac{\gamma_k}{\gamma} \mathbb{E}\tilde{g}_{i,k}^t \right\|^2$$

$$+ \frac{4\alpha^3 L^2 (1-2\gamma)^2}{\gamma^2} \mathbb{E}_t \left\| \frac{1}{N} \sum_{i \in N} \sum_{k=0}^{K-1} \frac{\gamma_k}{\gamma} \tilde{g}_{i,k}^t \right\|^2 + 56\alpha\eta_l^2 L^2 K (3\sigma_l^2 + 16\sigma_g^2 + 5L^2\rho^2)$$

$$\overset{(1)}{=} \mathbb{E}_t f(z^t) - \left( \frac{\alpha}{2} + \alpha\beta - 1064\alpha\eta_l^2 L^2 K - \alpha\beta^2 - L\alpha^2\beta^2 \right) \mathbb{E}_t \|\nabla f(z^t)\|^2$$

$$+ \left( \frac{4\alpha^3 L^2 (1-2\gamma)^2}{N^2\gamma^2} + \frac{L\alpha^2}{N^2} - \frac{\alpha}{2N^2} \right) \mathbb{E}_t \left\| \sum_{i \in N} \sum_{k=0}^{K} \frac{\gamma_k}{\gamma} \tilde{g}_{i,k}^t \right\|^2 + 56\alpha\eta_l^2 L^2 K (3\sigma_l^2 + 16\sigma_g^2 + 5L^2\rho^2)$$

$$
+ \alpha L^2 \rho^2 + \frac{2\alpha^3 L^2 (1-2\gamma)^2}{\gamma^3} \left( \mathbb{E}_t \left\| \frac{1}{N} \sum_{i \in N} h_i^t \right\|^2 - \mathbb{E}_t \left\| \frac{1}{N} \sum_{i \in N} h_i^{t+1} \right\|^2 \right)
$$

$$
+ \frac{56\alpha L^2 \eta_l^2 K}{\gamma N} \sum_{i \in N} \left( \mathbb{E}_t \|h_i^t\|^2 - \mathbb{E}_t \|h_i^{t+1}\|^2 \right). \tag{30}
$$

(2) holds due to the fact $\beta \in (0,1)$. We set $\alpha$ to satisfy $\frac{4\alpha^3 L^2 (1-2\gamma)^2}{N^2 \gamma^2} + \frac{L\alpha^2}{N^2} - \frac{\alpha}{2N^2} \leq 0$ and we make $\alpha\omega = \frac{\alpha}{2} + \alpha\beta - 896\alpha\eta_l^2 L^2 K - 168\alpha\eta_l^2 L^2 K\beta^2 - \alpha\beta^2 - L\alpha^2\beta^2$, and $\omega$ can be regarded as a constant.

*proof for $\omega$ can be regarded as a constant.* First, Let $\beta = 0$ means no dual variable correction exists. There exist a constant $c \in (0,1/2)$, we let $\omega = \frac{1}{2} - 1064\eta_l^2 L^2 K \geq \frac{1}{2} - c > 0$. Thus, $\omega = \frac{1}{2} - 1064\eta_l^2 L^2 K \geq \frac{1}{2} - c$ when the $\eta_l \leq \frac{\sqrt{c}}{\sqrt{1064 K L}} < \frac{1}{\sqrt{2128 K L}}$. For the final convergence, $\frac{1}{\omega} \leq \frac{2c}{1-2c}$ is a constant upper bound. When $\beta$ satisfy $\beta \leq \frac{1}{1+L\alpha}$, the upper bound on $\frac{1}{w}$ does not changed. $\qquad \square$

We take the full expectation on the bounded global gradient as:

$$
\alpha\omega \mathbb{E}\|\nabla f(z^t)\|^2 \leq \left( \mathbb{E}f(z^t) - \mathbb{E}f(z^{t+1}) \right) + \frac{56\alpha L^2 \eta_l^2 K}{\gamma N} \sum_{i \in N} \left( \mathbb{E}\|h_i^t\|^2 - \mathbb{E}\|h_i^{t+1}\|^2 \right)
$$

$$
+ \frac{2\alpha^3 L^2 (1-2\gamma)^2}{\gamma^3} \left( \mathbb{E}_t \left\| \frac{1}{N} \sum_{i \in N} h_i^t \right\|^2 - \mathbb{E}_t \left\| \frac{1}{N} \sum_{i \in N} h_i^{t+1} \right\|^2 \right)
$$

$$
+ 56\alpha\eta_l^2 L^2 K (3\sigma_l^2 + 16\sigma_g^2 + 5L^2\rho^2) + \alpha L^2 \rho^2. \tag{31}
$$

Take the full expectation and telescope sim on the above inequality:

$$
\frac{1}{T} \sum_{t=0}^{T-1} \mathbb{E}\|\nabla f(z^t)\|^2 \leq \frac{1}{T\alpha\omega} \left( f(z^0) - \mathbb{E}_t f(z^T) \right) + \frac{56\alpha L^2 \eta_l^2 K}{T\gamma N\omega} \sum_{i \in N} \mathbb{E}\|h_i^0\|^2
$$

$$
+ \frac{2\alpha^2 L^2 (1-2\gamma)^2}{T\gamma^3 \omega} \mathbb{E}_t \left\| \frac{1}{N} \sum_{i \in N} h_i^0 \right\|^2
$$

$$
+ \frac{1}{\omega} \left( 56\eta_l^2 L^2 K (3\sigma_l^2 + 16\sigma_g^2 + 5L^2\rho^2) + L^2\rho^2 \right). \tag{32}
$$

Here, we summarize the conditions and some constraints in the above conclusion. Like Sun et al. (2023c), we note that $(1 - (1 - \eta_l/\alpha)^K) < 1$ when $\eta_l \leq \alpha$. we have $1/\gamma > 1$. Whem $K > \alpha/\eta_l$, $(1 - \frac{\eta_l}{\alpha})^K \leq e^{-\eta_l K/\alpha} \leq e^{-1}$, then, $\gamma > 1 - e^{-1}$ and $\frac{1}{\gamma} < \frac{e}{e-1} < 2$. And apply the fact $f^* \leq f(x), \quad \forall x \in \mathbb{R}^d$:

$$
\frac{1}{T} \sum_{t=0}^{T-1} \mathbb{E}\|\nabla f(z^t)\|^2 \leq \frac{1}{T\alpha\omega} \left( f(z^0) - f^* \right) + \frac{112 L^2 \eta_l^2 K}{TN\omega} \sum_{i \in N} \mathbb{E}\|h_i^0\|^2
$$

$$
+ \frac{16\alpha^2 L^2}{T\omega} \mathbb{E}_t \left\| \frac{1}{N} \sum_{i \in N} h_i^0 \right\|^2 + \frac{1}{\omega} \left( 56\eta_l^2 L^2 K (3\sigma_l^2 + 16\sigma_g^2 + 5L^2\rho^2) + L^2\rho^2 \right).
$$

$$
\tag{33}
$$

This completes our proof of the Theorem 2.

