# OpenReview forum: "Neighborhood and Global Perturbations Supported SAM in Federated Learning:  From Local Tweaks To Global Awareness"
_ICLR.cc/2025/Conference — ICLR 2025 Conference Withdrawn Submission_

### Official Review · Reviewer_PARN · 2024-10-21

**Soundness:** 3
**Presentation:** 2
**Contribution:** 3
**Rating:** 5
**Confidence:** 4

**Summary:**

This paper proposes an improved optimizer based on SAM, which further enhances the global perspective when applying SAM in federated learning (FL) and maintains global generalization. Both the theoretical analysis and extensive experiments confirm the effectiveness of the proposed method.

**Strengths:**

1. The design motivation of this paper is reasonable. Although there are some flaws in the writing, the overall structure is understandable.

2. The experiments are very thorough, with extensive empirical studies conducted under standard settings to validate the efficiency of the proposed techniques.

3. A convergence analysis was conducted for the proposed method to demonstrate that its convergence remains at the same level as the previous works.

**Weaknesses:**

1. The section on Methods (Section 4) is quite disorganized. I suggest that the authors include a notation table to explain all the variables that appear later in the text. I noticed that many variables are introduced without explanation, see questions.

2. The definition of Equation (9) seems somewhat obscure and difficult to understand. The vanilla local objective (like FedDyn) adopts the the augmented alternating direction method of multipliers to solve the consensus problem. The term $h_i$ is the dual variable to balance the dual problem. Performing operations on the dual term seems to affect the consistency solution of the primal problem. I suggest that the authors remove the problem formulation related to Equation (9) and directly introduce the use of certain variables to replace or correct the gradient. Actually, from the optimization perspective, the proposed method still solve eq.(8), but with the novel proposed method (momentum-based gradient estimator and SAM-based local optimizer). If the authors modify the entire Lagrangian function, it still need to be proven that the solution of this function is consistent with the solution of the original problem. I believe this is unnecessary for the techniques proposed in this paper.

**Questions:**

1. A technical question: in line 245, according to the motivation of FedCM that $\Delta^t\approx \nabla f(\theta^t)$, why local perturbation is not $\delta_k^t=\rho\frac{(1-\kappa)g_{i,k}^t + \kappa\Delta^t}{\Vert (1-\kappa)g_{i,k}^t + \kappa\Delta^t \Vert}$? It looks like a external momentum in current version. Will the current external momentum setting outperform the original inner momentum form estimation of FedCM?

2. I am confused on line 264. What is the term $g_{i,k}$? Why the fusion term performs as $g_{i,k} + \widetilde{g}_{i,k-1}^t + \kappa\Delta^t$?

3. Line 11 in Algorithm 1, what is the term $g_{i,k}^{t}[\widetilde{g}_{i,k-1}^t]$?

4. What is the conection between the term in line 264 and "lookahead" optimizer? Although the paper claims that this form is an extension of lookahead, the authors could provide the corresponding extended formulas to further explain why this form corresponds to lookahead optimizers.

5. Could the authors compute the variance of the SAM perturbations? In fact, if the global perturbation is approached more closely, the variance of their local perturbations should be smaller. Additionally, the authors could calculate the global perturbation at the beginning of each round and compare whether this approach leads to further improvements.



Some typos:

(1) There are issues with the references in this paper. I suggest that the authors distinguish between the \citet and \citep commands to provide correct citations.

(2) Line 901 "waht" to what

---

> ### Author Response · Authors · 2024-11-13
>
> Dear Reviewer PARN,
> We are grateful for your constructive feedback.
>
> Weakness 1: We greatly appreciate your comments. To address concerns regarding the use of symbols, we have added a table of variable explanations in the appendix.
>
> Weakness 2: Your suggestions have been highly beneficial to us. We have decided to eliminate the description related to Eq. 9 to maintain the original expression of AL while introducing the global gradient as a direct technical means in the article. We believe these adjustments will enhance the clarity and coherence of our work.
>
> Question 1: Regarding the consideration of not treating the global gradient as a weight factor in the perturbation calculation, we have made the following deliberations. We aim to preserve the integrity of local gradients because we have observed that with the progression of training in FedCM and MoFedSAM, the advantages in accuracy and convergence speed gradually diminish. We hypothesize that this may be due to the trade-off with the global gradient, which hinders the model's ability to effectively learn local knowledge.
>
> Questions 2-3: We apologize for any confusion caused by our textual presentation. Firstly, let us clarify that "g_i^k" refers to an unbiased estimate of "\nabla f_i(\theta_i,\xi_i)", while "\tilde{g}_i^k" denotes the gradient calculated after model perturbation. The key aspect of our proposed neighborhood perturbation is its effective utilization of historical gradients stored in the gradient cache. Since the gradient cache needs to be cleared before each training session, and SAM performs a gradient clear operation before integrating gradient ascent, by disabling this operation, we can retain the previous gradient, thus effectively integrating it into the current SAM computation, leading to a fused computation method. After the SAM computation, the gradient cache is cleared again without interfering with subsequent SGD calculations. For a more detailed observation of the changes in the gradient cache, please refer to Appendix A.3.
>
> Question 4: This scheme is not an extension of LookAhead (though it draws inspiration from it). Specifically, given the computation method of SAM, we consider incorporating the gradient from the previous update step into the gradient ascent process similar to the gradient lookback operation in LookAhead. Therefore, we integrate this scheme with existing methods to validate the effectiveness of neighborhood perturbation. Detailed experiments can be found in Appendix B.7.
>
> Question 5: We have promptly included the model drift experiment in the appendix under the experimental settings of LeNet on CIFAR10.
>
> We understand your concerns, and we sincerely hope that our responses will fully address them and contribute to the further improvement of our manuscript. If you have any additional questions or require further clarification, please do not hesitate to let us know.

---

### Official Review · Reviewer_agaF · 2024-10-31

**Soundness:** 2
**Presentation:** 2
**Contribution:** 3
**Rating:** 5
**Confidence:** 3

**Summary:**

To address the local optima divergence in Heterogeneous Federated Learning, this paper proposes a FedTOGA method by linking the local dynamic regularizer to global updates to enhance the consistency of optimization and generalization. The method efficiently links local perturbations to global updates and achieves a non-convex convergence rate of $\mathcal{O}(1/T)$. The authors also propose neighborhood perturbation to approximate local perturbation. The authors also provide empirical validations of the theoretical results as well.

**Strengths:**

1. The proposed method is well-motivated, the paper shows that existing methods suffer from the local optima divergence issue, and show how to fix it.
2. The empirical results show that the FedTOGA method is better than other HtFL methods using global sharpness-aware minimization (SAM) and dynamic regularization, as expected.

**Weaknesses:**

1. In Theorem 2, compared with other methods using global sharpness-aware minimization (SAM) and dynamic regularization, such as [1],  a more distinct summary of the superiority of FedTOGA may be needed.
2. In addition to CIFAR10/100, the authors should also consider the performance of FedTOGA in the TinyImageNet task.
3. In FL research, the local epoch is an important parameter. The authors should study this parameter's impact on performance.

**Questions:**

See in weaknesses.

---

> ### Author Response · Authors · 2024-11-13
>
> Dear Reviewer agaF,
>
> We sincerely appreciate your constructive feedback, which has greatly contributed to the improvement of our manuscript.
>
> Weakness 1:
> We appreciate the reviewers' feedback and would like to clarify that our approach with FedTOGA no longer presupposes that local training for each client must reach a stable equilibrium. Unlike FedSpeed, which adheres to a more rigid framework, FedTOGA introduces a unique perturbation mechanism that effectively relaxes this constraint, allowing for more dynamic and flexible local training processes.
>
> Furthermore, by adjusting the value of \beta, we can precisely control the $1/omega$ ratio, thereby optimizing the balance between local and global model updates. This adjustment is a key feature that enhances the adaptability and performance of our federated learning system.
>
> Weakness 2: To address the reviewers' concerns, we have initiated testing on the TinyImageNet dataset. In the previous version, we adhered to the standard settings used in FedSMOO and FedLESAM to ensure fair and accurate benchmarking of all algorithms. Due to space limitations, we omitted these results in the earlier submission.
>
> Weakness 3: In the appendix, we have included a detailed critical analysis of the local training rounds K. This analysis explores the impact of different K values on model performance, providing deeper insights into the optimization process.
>
> We understand the importance of addressing the reviewers' concerns and believe that these additions will significantly enhance the clarity and robustness of our paper. If you have any further questions or require additional information, please feel free to contact us.

---

### Official Review · Reviewer_fUrX · 2024-11-03

**Soundness:** 2
**Presentation:** 1
**Contribution:** 2
**Rating:** 3
**Confidence:** 4

**Summary:**

The paper proposes a global-aware perturbation method for sharpness-aware minimization (SAM) in federated learning (FL). Compared with existing methods, this paper combines the idea of local dynamical regularizer with global updates to mitigate the effect of data heterogeneity. Convergence of the proposed algorithm is theoretically analyzed with the rate $O(1/T)$. Empirical results are shown to justify the performance of the algorithm.

**Strengths:**

The main contribution of the paper is to propose a local and global perturbation-based algorithm (FedTOGA) that enjoys communication and computation efficiency for SAM in FL. Both theoretical convergence guarantee and empirical evaluation of the algorithm are provided.

**Weaknesses:**

1. The novelty and contribution of the paper is limited. I think the main advantage of the proposed algorithm that authors claim is to leverage the global and neighborhood perturbation to reduce communication overhead and computational cost. However, this is not clear in the paper. As in Theorem 2 the authors claim that the rate of $O(1/T)$ is achieved when setting $K=O(T)$, which is faster than existing literature. However, FedSMOO also achieves the same rate $O(1/T)$ and linear speedup in the number of clients without any constraint on $K$. Thus, the advantage of FedTOGA is not clear.

2. The analysis tools look quite similar to those in FedSpeed paper. Thus, the technical contribution of this paper seems limited.

3. The presentation of the paper is not good, which makes the reader hard to follow. There are some mistakes and typos. To list a few, in Line 300,  there is no "Line 16" in the algorithm. In Theorem 2, $z_t$ is not defined and in eq. (11) the LHS should be the sum of $\Vert z_t \Vert^2$.

**Questions:**

1. Could the authors further explain why their algorithm is better comparing to previous methods in terms of e.g. convergence rate, computational cost, challenge in implementation, etc?

2. What are the technical difficulties of the theoretical analysis, comparing to literature?

3. As in Theorem 2, the choice of learning rate is with order $O(1/\sqrt{T})$ while constants are ignored. However, these neglected constant may significantly influence the actual performances of the algorithms. Could the authors explain why the learning rates are identical for all algorithms? In the above sense, does it cause a fair comparison? If yes, could the authors explain the reason why this renders a fair comparison?

---

> ### Author Response · Authors · 2024-11-13
>
> Dear Reviewer fUrX,
> We are very grateful for your constructive comment.
> Weakness 1: Thank you for your valuable comments on our paper. Firstly, we would like to clarify that our primary contribution is not to introduce an entirely new theory but rather to propose a novel global SAM estimation method and the introduction of dynamic regularization for global updates. Additionally, we innovatively introduce the technique of neighborhood perturbation. For a deeper understanding of the limitations of existing algorithms, such as FedSMOO and FedLESAM, please refer to Appendix A.2. Notably, these methods universally require additional local storage space for computation. In contrast, FedTOGA passively receives global updates from the server and integrates them effortlessly, significantly reducing the requirements for stable client connections and storage space.
>
> Weakness 2: Thank you for your insightful comments. "Standing on the shoulders of giants," we acknowledge that our convergence analysis is based on FedSpeed. Unlike FedSMOO and FedDyn, which assume that local clients stop at a local stable point after each training round—an assumption that is almost impossible in federated learning (FL)—our approach does not rely on this assumption. FedSpeed extends the analysis to K local iterations but imposes more restrictions on the perturbation learning rate. FedTOGA introduces a different perturbation mechanism compared to FedSpeed, thereby relaxing these restrictions. Adjusting the value of  \beta can further tighten $1/omega$. We have also corrected some errors in the analysis process of FedSpeed. While our method may not show significant advantages in convergence speed, it is more broadly applicable compared to the analyses of FedSMOO and FedSpeed.
>
> Weakness 3: Thank you for your valuable feedback. We have thoroughly revised the manuscript to improve clarity and correctness, addressing the issues with presentation, typos, the missing line in Algorithm 1, undefined terms in Theorem 2, and the error in Equation (11). We hope these changes enhance the readability and accuracy of our paper.

---

### Official Review · Reviewer_gca5 · 2024-11-05

**Soundness:** 3
**Presentation:** 2
**Contribution:** 2
**Rating:** 3
**Confidence:** 3

**Summary:**

This paper presents FedTOGA, a Federated Learning (FL) algorithm  designed to prevent the heterogeneity of client data cause the global model to converge to a sharp local minimum. FedTOGA achieves this in a
communication-efficient way by combining new variants of two techniques: (i) Sharpness-Aware Minimization (SAM) to add perturbations to the training process and (ii) local dynamic regularization. In contrast to existing literature, FedTOGA uses the global gradient update to adjust both the global perturbation and the local regularization. The authors obtain analytic convergence guarantees for their methodology, and show the effectiveness of their approach by conducting extensive experiments.

**Strengths:**

+ Comprehensive validation: In Sec. 6, FedTOGO is compared against a lot of FL algorithms and is able to outperform all of them.

**Weaknesses:**

+ In the first contribution bullet point in Sec. 1, the claim about FedTOGA being the first global perturbation technique and first local dynamic regularizer needs to be rephrased to emphasize more on the fact that it is the first to do so using the global update. The current version of this statement overlooks the contributions of FedSMOO, FedLESAM, FedSpeed and papers that use dynamic regularizers, which have developed these ideas but without using global updates.
+ In Sec. 2's Sharpness-Aware Minimization, the parameters \rho and \delta need to clearly defined.
+ In Eq. (4) given in Sec. 2.1, the notation \theta_i seems to imply the different local models for clients in the FL setup. If this is the case, this needs to be clarified by defining \theta_i, which is missing currently. The same comment applies for Eq. (6) in Sec. 4.1, where the minimization is being done over a single vector \theta while there are multiple \theta_i's in the loss function formulation.
+ In Assumption 3 in Sec. 5, can the authors explain why they need a bounded variance of the unit gradient, and why just a bounded variance of the gradient itself is not sufficient? Adding some references which also make this assumption for the unit gradient is recommended.
+ In Sec. 6, does FedTOGO and most of the other FL algorithms perform similarly if the data distribution among clients is IID? Currently all experiments are done in non-IID settings and it would be insightful to
see how these FL methods compare in IID setups.
+ The global perturbation used in this paper build heavily on prior research, doing a gradient ascent similar in existing algorithms with the addition of some global update information. The same goes for the local regularizer. While it is interesting to see how much accuracy increase can be achieved by making these adjustments, I am concerned about the novelty of this incremental idea. Alongside other issues mentioned above, I do not think this paper is ready to be published at ICLR in its current version.

**Questions:**

See weaknesses.

---

> ### Author Response · Authors · 2024-11-13
>
> Dear Reviewer gca5, We are deeply appreciative of your constructive comment.
>
> Strengths: We are grateful for your encouragement. In future versions (subject to submission size limitations), we will provide a thoroughly segmented dataset to facilitate the reproducibility of our work by all researchers. Additionally, all comparative baseline results can be referenced from FedSMOO, which will further support the reproducibility of our study.
>
> Weakness 1: We appreciate your feedback and recognize that there may have been an issue with our articulation. Our contribution primarily emphasizes being the first to introduce global updates into SAM and dynamic regularization, which, to the best of our knowledge, has not been addressed in previous studies. We certainly do not intend to overlook the significant contributions of FedSMOO, FedLESAM, and FedSpeed regarding the use of dynamic regularization. Instead, we have repeatedly acknowledged these contributions throughout the main text and appendices. For example, see Abstract lines 015-016, Introduction lines 073-075, Appendices A.1 and A.2, as well as the Contributions section. Additionally, we revisit the limitations of dynamic regularization in prior research in Section 2.2. This combination is innovative, and to our knowledge, no study has mentioned this ingenious integration.
>
> Weakness 2: The parameter "\rho" refers to the perturbation learning rate, and "\delta" denotes the global update gradient. We have added more detailed explanations in the main text to clarify these points.
>
> Weakness 3: "theta_i" represents the local model loaded on the client side. We have further refined the notation explanations and added a table of variable definitions in the appendix to enhance clarity.
>
> Weakness 4: The introduction of bounded variance for unit gradients is aimed at facilitating the derivation of Theorem 1, which demonstrates that generalization error and optimization error are geometrically amplified as the local interval expands. This assumption is crucial for supporting the rationale behind our motivation (Section 3). It is also consistent with the assumptions made in related works such as FedSAM and FedLESAM.
> [1]. Qu Z, Li X, Duan R, et al. Generalized federated learning via sharpness aware minimization[C]//International conference on machine learning. PMLR, 2022: 18250-18280.
> [2]. Fan Z, Hu S, Yao J, et al. Locally Estimated Global Perturbations are Better than Local Perturbations for Federated Sharpness-aware Minimization[C]//Forty-first International Conference on Machine Learning.
>
> Weakness 5: We appreciate your concern regarding the IID testing. In response, we have initiated comprehensive IID testing to address this issue. Specifically, we show that by adjusting the hyperparameters, the performance of FedTOGA can be restored to its original version, ensuring that it will not perform worse than existing algorithms.
> Furthermore, we would like to emphasize that our comparison scenarios strictly adhere to the Non-IID experimental settings used in FedSMOO, FedSpeed, and FedLESAM. This alignment with established experimental protocols allows us to compare our results directly with the original experimental data, thereby ensuring the reliability and validity of our findings.
>
> Weakness 6: Thank you for your valuable comments on our paper. "Standing on the shoulders of giants," our algorithm is indeed inspired by FedSMOO, FedCM, FedSpeed, and FedLESAM. We acknowledge that it is challenging to reasonably integrate existing technologies and provide further explanations. However, rethinking and combining established techniques from a completely new perspective is highly significant. Most importantly, our integration significantly reduces the resource requirements of local clients in both FedSMOO and FedLESAM while achieving better results.
> In addition, we have introduced the novel concept of  Neighborhood perturbation for the first time. This innovative use of gradient caching optimizes the computation of perturbations, a solution that can be effectively applied to existing SAM-based algorithms. For more details, please refer to Appendix B.7. After implementing this technology, there has been a significant improvement in the convergence speed and accuracy of FedSpeed and FedSMOO.
>
> We would also like to kindly remind you that the name of our algorithm is FedTOGA, not FedTOGO.
>
> We appreciate your concerns and hope that our response adequately addresses them and contributes to the further refinement of our manuscript. Should you have any additional questions or require further clarification, please do not hesitate to let us know.

---

### Author Response · Authors · 2024-11-16
**General Response**

We are eager for further discussions to elucidate this paper's contributions and correct relevant errors, and we thank the reviewers for their valuable time. Most studies go a step further based on existing studies. Please refer to Table 5 to see how we differ from existing studies. Once again, we would like to highlight our proposed $\textbf{neighborhood perturbation}$ technique, which is a completely new attempt in FL.

---

### Note · Authors · 2024-11-16

I have read and agree with the venue's withdrawal policy on behalf of myself and my co-authors.